# Estrogenic endocrine disruptor exposure directly impacts erectile function
Samuel M. Cripps[1], Sarah A. Marshall[2], Deidre M. Mattiske[1], Rachel Y. Ingham[1] & Andrew J. Pask [1]✉

Erectile dysfunction (ED) is an extremely prevalent condition which significantly impacts quality of life. The rapid increase of ED in recent decades suggests the existence of unidentified environmental risk factors contributing to this condition. Endocrine Disrupting Chemicals (EDCs) are one likely candidate, given that development and function of the erectile tissues are hormonally dependent. We use the estrogenic-EDC diethylstilbestrol (DES) to model how widespread estrogenic-EDC exposure may impact erectile function in humans. Here we show that male mice chronically exposed to DES exhibit abnormal contractility of the erectile tissue, indicative of ED. The treatment did not affect systemic testosterone production yet significantly increased estrogen receptor α (*Esr1*) expression in the primary erectile tissue, suggesting EDCs directly impact erectile function. In response, we isolated the erectile tissue from mice and briefly incubated them with the estrogenic-EDCs DES or genistein (a phytoestrogen). These acute-direct exposures similarly caused a significant reduction in erectile tissue contractility, again indicative of ED. Overall, these findings demonstrate a direct link between estrogenic EDCs and erectile dysfunction and show that both chronic and acute estrogenic exposures are likely risk factors for this condition.

Erectile dysfunction (ED) is defined as the chronic inability to gain or sustain an erection sufficient for satisfactory sexual performance[1]. The condition can range in severity [2] and significantly impacts quality of life for the afflicted individual and their partner[3]. In addition, ED can be an indicator for cardiovascular disease, a leading cause of death globally[4]. The exact prevalence of ED is difficult to determine as this relies on self-reporting and subjective data-gathering[5]. Nonetheless, in proportion to global population projections, ED cases worldwide were projected to increase from 2.66% in 1995 to 3.94% in 2025[6–9]. The global trend is reflected in the increasing percentage of ED cases of >65 year old men in the US Medicare population; from 2.32% in 2009 to 2.64% in 2015[10,11]. Furthermore, expenditure on ED treatment in the US rose from $185 million in 1994 to $330 million in 2000 (not including pharmaceutical costs)[5]. Given that ageing is incontrovertibly linked to ED[6,12–17], a global increase in the ageing population can partially explain these trends[8]. However, among younger men there is an exceptionally high prevalence and consulting for ED[17–21], demonstrating that ageing is not always a determinant for this condition. Overall, the rapid rise of ED globally has prompted further investigation and revision of ED aetiology.

Until the 1970s, psychogenic factors such as depression and anxiety [22] were thought to primarily contribute to ED. However, we now know that ED is associated with a range of physiological or organic factors including neurological, vascular, and endocrine abnormalities[23]. These factors are also associated with ageing, explaining its role in this condition[24]. Lifestyle factors such as smoking, poor diet and lack of exercise also have well-established links with ED[13,16,25–27], highlighting the impact environmental elements can have on erection physiology, the central theme of the present study.

The process of erection is mediated by three cylindrical structures in the adult penis which comprise the erectile tissue: the paired corpora cavernosa (CC) dorsal to the urethra and the smaller corpus spongiosum which encloses the urethra and forms the glans distally[28]. The CC are primarily responsible for erection and are a meshwork of sinusoidal spaces lined by smooth muscle and endothelium, which in turn are connected to the systemic blood supply via larger arteries[29]. Upon sexual stimulation, nitric oxide (NO) is released in the endothelium and nerve terminals innervating the CC by several mechanisms, which include release of the neurotransmitter acetylcholine (ACh) from cholinergic nerve terminals in the CC[30]. ACh interacts with its receptors on endothelium (thus, is endothelium-dependent) to stimulate endothelial NO production[31–33]. NO then diffuses directly into smooth muscle to stimulate cyclic guanosine 3′,5′-cyclic monophosphate (cGMP) production. In turn, cGMP expels cytosolic $Ca^{2+}$ by altering ion channel activity, causing smooth muscle relaxation[34–37]. As a result, the arteries and sinusoidal spaces of the CC engorge with blood

[1]School of BioSciences, The University of Melbourne, Melbourne, Australia. [2]The Ritchie Centre, Department of Obstetrics & Gynaecology, Monash University, Melbourne, Australia. ✉e-mail: ajpask@unimelb.edu.au

and dilate, causing erection[29]. Following sexual activity, the release of several factors within the CC including the neurotransmitter noradrenaline (NA), the prostanoid thromboxane $A_2$ (TxA$_2$) and the peptide endothelin-1 (ET-1) return cytosolic $Ca^{2+}$ to normal levels in smooth muscle cells, which in turn facilitates contraction and thus reversion to the flaccid penis[38]. Overall, the process of erection is primarily a vascular event, emphasising the strong link between ED and cardiovascular disease.

The patterning, maintenance, and physiology of the CC are all tightly controlled by hormones. In particular, androgens such as testosterone are critical in their positive regulation of CC nerve structure and density[39–42], smooth muscle enrichment[43–47], and smooth muscle cell ion channels[48,49]. In addition to androgens, the penis of the human, rabbit, rat, and mouse all express aromatase and the estrogen receptors (ER α and β)[50–58]; proteins which mediate estrogen synthesis and signal transduction, respectively. Endogenous estrogen potentiates smooth muscle contraction mediated by oxytocin signalling in the rabbit CC[54] and has a critical role in mouse penis development[59,60]. Overall, endogenous androgens and estrogens have profound effects on the physiology, patterning, and development of the male erectile tissue. Therefore, exposure to endocrine disrupting chemicals (EDCs) may be an important, yet unidentified, additional contributing factor to ED.

EDCs are defined as exogenous substances or mixtures which alter functioning of the endocrine system to produce adverse health outcomes in an organism, its progeny or (sub) populations[61]. EDC use has increased exponentially over the past few decades resulting in their ubiquity in the environment[62] and correlates with increases in ED prevalence. Furthermore, EDCs are already implicated in the rapid increase of male differences in sexual development (DSDs) such as cryptorchidism and hypospadias[63]. Although EDCs are known to impact the hormonal pathways which influence penis development and erection physiology[64], their role in ED aetiology is unclear, particularly regarding those which mimic estrogens (estrogenic-EDCs).

Estrogenic-EDCs are among the most pervasive and persistent chemicals in our environment and include bisphenols, pharmaceuticals and naturally occurring phytoestrogens (i.e., plant-derived)[65]. Estrogenic-EDC exposure typically induces adverse outcomes by targeting ERs localised in a target organ or tissue[66]. In males, this may also include activation of ERs in the Leydig cells, which in turn can reduce testosterone production from the testes[67]. Therefore, in males, estrogenic-EDC exposure may induce adverse effects by stimulating a local endogenous estrogen signalling system directly and/or eliciting an anti-androgenic effect leading to systemic testosterone deficiency. One of the best studied estrogenic-EDCs is the potent pharmaceutical diethylstilbestrol (DES), which activates the ER at an affinity similar to that of estradiol, a native estrogen[68]. A potential link with ED is demonstrated by DES-treated neonatal rats which exhibit severe disruption of CC development and morphology[69–73]. Historically, DES was mistakenly believed to prevent miscarriage and it is estimated that between the 1940–70s, DES was prescribed to 10 million pregnant women before it was banned in 1971 due to the resulting adverse health effects[74]. In male DES descendants, these included DSDs such as cryptorchidism as well as a slight increased risk of infertility[75,76]. In addition to historical exposure, current patients with prostate cancer may receive DES supplements to reduce testosterone levels[77]. Here, DES is used to model widespread exposure to the multitude of estrogenic-EDCs across potentially sensitive developmental and adult timepoints.

To provide more definitive evidence for a potential link between estrogenic-EDCs and ED, this study also tests the effects of the estrogenic-EDC genistein on erectile function. Genistein is a phytoestrogen (i.e., a plant-derived estrogen) which is abundant in soy-based products compared to other legumes[78]. Excessive soy (genistein) consumption is reported in case studies as an independent causal factor for ED[79–81], suggesting that genistein is a contributing factor. Genistein exposure can also cause reduced testosterone levels, suggesting estrogenic-EDCs contribute to ED indirectly by functioning as an anti-androgen or by direct activation of ERs localised in the CC, or both[70–72,79,81]. Although genistein is defined as an estrogenic-EDC, its biological effects also include estrogen-independent pathways which may also impact erectile function[82,83]. Taken together, there is emerging evidence that estrogenic-EDC exposure is a risk factor for ED which prompts further investigation. However, our current understanding is restricted by the limited number of studies which have functionally tested the effects of estrogenic-EDC exposure on the CC[84–86].

The aim of this study was to determine whether EDC exposure can directly impact erection physiology. Here, we investigate the effects of systemic DES exposure in male mice via drinking water. Treatment occurred in utero and postnatally, when it is predicted to impact patterning of the CC[87,88], as well as during adulthood. Following treatment, we analysed ex vivo function of the CC at the adult stage, as well as its gene expression and morphology. In contrast to systemic estrogenic-EDC exposure, no study has tested the effects of direct estrogenic-EDC exposure on CC function. DES is a potent activator of ER and thus may interfere with endogenous estrogen signalling localised to the CC, constituting a direct mechanism of action. However, estrogenic-EDCs are also known to reduce testosterone production from the testes. Given that testosterone is critical for CC development and function, this represents a potential indirect mechanism by which estrogenic-EDCs contribute to ED. To better understand the underlying mechanism(s) and impact of estrogenic-EDC exposure on erectile function, it is important to disentangle their direct and indirect effects. Therefore, we isolated CC samples from untreated mice and performed short-term incubations in DES and genistein, followed by analysis of gene expression and function ex vivo. Finally, since ED is an established indicator for cardiovascular disease in humans, we investigated whether EDCs could impact mesenteric artery function to identify their potential for broad cardiovascular impacts.

## Results

### Systemic DES exposure did not impact androgen levels

The DES$_{water}$ mice exhibited no significant alterations to AGD, testes weights, seminal vesicle weights or plasma testosterone levels (Fig. 1). The corresponding numerical values are available in Supplementary Data 1.

### Systemic DES treatment amplified contraction responses in CC without altering erectile tissue histology

To assess whether a chronic DES exposure affected relaxation of the CC, response curves of the CC were generated in the presence of either an endothelium-dependent (ACh) or -independent (SNP) dilator. ACh induced relaxation in the DES$_{water}$ and control CC, designated here as the '1$^{st}$ Phase' of the ACh response. Conversely, as the concentration of ACh increased, the CC contracted rather than reach a baseline; designated the '2$^{nd}$ Phase' of the ACh (Fig. 2a). DES$_{water}$ CC samples did not exhibit significant differences in ACh sensitivity, overall relaxation, or maximum relaxation (Suppl. Table 1). The DES$_{water}$ CC samples exhibited a significantly higher contraction level in the 2$^{nd}$ phase (ACh [10 μM] relaxation %; control: 49.3 ± 4.73; DES$_{water}$: 33.3 ± 5.93; $P = 0.04$) (Fig. 2b). SNP induced relaxation in the DES$_{water}$ and control CC which showed no significant differences for sensitivity, overall relaxation, and maximal relaxation (Fig. 2c; Suppl. Table 1).

After assaying for relaxation, we next assessed whether a systemic DES exposure modified the ability of the CC to contract. The thromboxane $A_2$ (TxA$_2$) receptor agonist, U46619, contracted CC samples and systemic DES exposure significantly increased sensitivity to this factor ($-logEC_{50}$ [U46619] M; control: 7.22 ± 0.06; DES$_{water}$: 7.48 ± 0.03; $P = 0.003$) with no significant differences for overall or maximal contraction (Fig. 3; Suppl. Table 1). Histological analysis demonstrated no significant difference in the smooth muscle to collagen ratio of the DES$_{water}$ mice (Fig. 4). The corresponding numerical values are available in Supplementary Data 1.

### Systemic DES exposure increased Esr1 expression in the CC

RT-PCR of DES$_{water}$ CC samples was used to assess expression of genes associated with steroid signalling, erection physiology, ED, and penis development. *Esr1* (encoding ERα) expression was significantly increased in

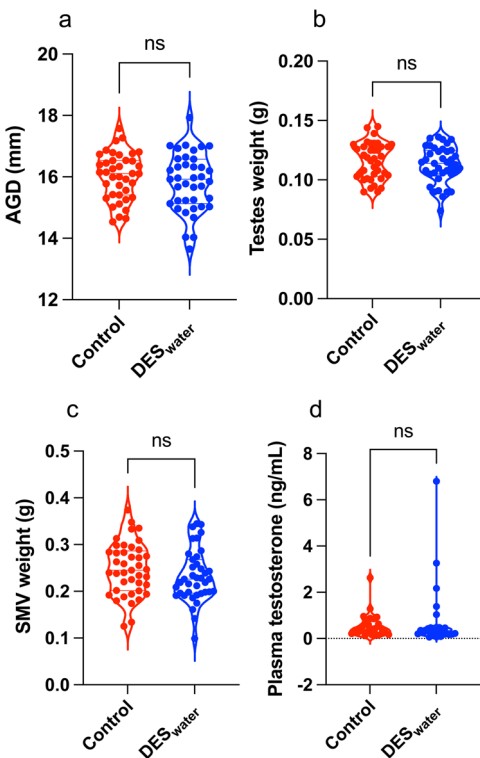

**Fig. 1 | Anogenital distance, testes weight, seminal vesicle weight and plasma testosterone in DES_water mice. a–d** Violin plots for parameters listed in figure caption title for DES_water (blue) and control (red) mice. **a** No significant difference in anogenital distance (AGD) (mm; control: $16.03 \pm 0.12$, $n = 39$; DES_water: $15.83 \pm 0.15$, $n = 39$; $P = 0.31$; unpaired, two-tailed t-test). **b** No significant difference in testes weight (g; control: $0.12 \pm 0.002$, $n = 39$; DES_water: $0.11 \pm 0.002$, $n = 39$; $P = 0.37$; unpaired, two-tailed t-test). **c** No significant difference in seminal vesicle (SMV) weight (g; control: $0.25 \pm 0.009$, $n = 41$; DES_water: $0.23 \pm 0.009$, $n = 39$; $P = 0.23$; unpaired, two-tailed t-test). **d** No significant difference in plasma testosterone concentration (ng/mL; control: $0.56 \pm 0.10$, $n = 27$; DES_water: $0.79 \pm 0.28$, $n = 26$; $P = 0.37$; Mann–Whitney test).

the DES_water CC normalized to the housekeepers *Rpl13a* (control: $0.88 \pm 0.10$; DES_water: $1.54 \pm 0.21$; $P = 0.02$) and *Rps29* (fold change; control: $0.82 \pm 0.06$; DES_water: $1.59 \pm 0.22$; $P = 0.01$) (Fig. 5a; Suppl. Table 2). No significant differences were observed for *Ar*, *Tbxar2*, *Sfrp1*, *Angpt4*, *Igfbp3*, *Nos3* and *Oxtr* normalized to *Rpl13a* (Fig. 5b–h). The corresponding numerical values are available in Supplementary Data 1.

### Acute direct DES exposure caused CC dysfunction

After confirming that a chronic, systemic exposure of DES enhances CC contraction and *Esr1* expression, we next assessed whether an acute direct exposure to DES or genistein (see below) would also alter CC function. Initially, a dose of 10 µM was used to assess the effect of DES (DES_direct-10). During the wire myography assay, the DES_direct-10 CC samples failed to maintain contractile tone which indicated severely dysfunctional tissue (Suppl. Fig. 1). Consequently, the DES_direct-10 CC samples could not be assayed for relaxation responses, as this relies on maintenance of contractile tone (see methods). After reducing the dose to 5 µM, the DES_direct-5 CCs did not exhibit the 2nd phase of contraction (observed only in the controls) and were significantly desensitized to ACh (−logEC50 [Ach] M; control: $7.31 \pm 0.06$; DES_direct-5: $6.61 \pm 0.08$; $P = 0.0001$) (Fig. 6a, b). No significant differences were observed for maximal relaxation or overall relaxation (Suppl. Table 1).

DES_direct-5 CC samples and the corresponding vehicle-treated controls relaxed to SNP (Fig. 6c). Maximal relaxation to SNP was significantly higher in the DES_direct-5 samples (R_max %; control: $72.88 \pm 7.23$; DES_direct-5: $93.15 \pm 5.36$; $P = 0.04$) (Fig. 6d). SNP sensitivity and overall relaxation were

unaltered (Suppl. Table 1). The corresponding numerical values are available in Supplementary Data 1.

### Acute direct DES exposure did not impact expression of genes associated with steroid signalling, endothelial function, apoptosis, or anti-apoptosis

We performed RT-PCR of genes involved in steroid signalling, endothelial function, apoptosis, and anti-apoptosis. Following normalization of DES_direct-5 CC cDNA content to *Rpl13a*, no significant differences in relative fold change were observed for *Ar*, *Esr1*, *Nos3*, *BclXL*, *Bcl2*, *Parp1*, *Bak1* and *Bax* (Fig. 7). The corresponding numerical values are available in Supplementary Data 1.

### Acute direct genistein exposure reduced CC sensitivity to endothelium-dependent relaxation factors

After confirming that acute direct DES exposure could impact CC function, we next tested the direct effects of the less potent estrogenic-EDC genistein on endothelium-dependent (ACh) and -independent (SNP) CC relaxation. The Gen_direct-20 CC samples and corresponding vehicle controls exhibited the 1st and 2nd Phases of the ACh response, although the 2nd Phase was more pronounced in the controls (Fig. 8a). In the ACh curve, the Gen_direct-20 CC samples were significantly desensitized (−logEC50 [ACh] M; control: $7.64 \pm 0.07$; Gen_direct-20: $7.34 \pm 0.09$; $P = 0.002$), displayed a trend for reduced maximum relaxation to ACh (R_max %, control: $106.6 \pm 8.36$; Gen_direct-20: $94.29 \pm 7.37$; $P = 0.05$), and were unaltered for overall relaxation to ACh (Fig. 8b; Suppl. Table 1). The Gen_direct-20 CC samples and corresponding controls all relaxed to SNP, with no statistical differences for sensitivity, overall relaxation, or maximal relaxation (Fig. 8c; Suppl. Table 1). The corresponding numerical values are available in Supplementary Data 1.

### Acute direct Genistein exposure reduced Nos3 expression in the CC

We performed RT-PCR of genes involved in steroid signalling, endothelial function, apoptosis, and anti-apoptosis. No significant differences were observed for expression of *Ar* and *Esr1* normalized to *Rpl13a* (Fig. 9a, b). *Nos3* was significantly decreased in Gen_direct-20 CC samples normalized to *Rpl13a* (control: $1.12 \pm 0.25$; Gen_direct-20: $0.51 \pm 0.08$; $P = 0.046$) and exhibited a corresponding decreased trend normalized to *Rps29* (control: $1.06 \pm 0.16$; Gen_direct-20: $0.67 \pm 0.12$; $P = 0.08$) (Fig. 9c; Suppl. Table 2).

The anti-apoptotic markers *BclXL* and *Bcl2* were significantly reduced in Gen_direct-20 CC samples normalized to *Rpl13a* (*BclXL*; control: $1.15 \pm 0.27$; Gen_direct-20: $0.52 \pm 0.07$; $P = 0.046$; *Bcl2*; control: $1.04 \pm 0.14$; Gen_direct-20: $0.51 \pm 0.07$; $P = 0.01$) (Fig. 9d, e). These markers displayed a corresponding decreased trend normalized to *Rps29* (*BclXL*; control: $1.14 \pm 0.23$; Gen_direct-20: $0.70 \pm 0.13$; $P = 0.13$; *Bcl2*; control: $1.05 \pm 0.16$; Gen_direct-20: $0.68 \pm 0.14$; $P = 0.12$) (Suppl. Table 2).

The apoptotic markers *Parp1* and *Bak1* were significantly reduced normalized to *Rpl13a* (*Parp1*; control: $1.22 \pm 0.38$; Gen_direct-20: $0.46 \pm 0.13$; $P = 0.02$; *Bak1*; control: $1.04 \pm 0.13$; Gen_direct-20: $0.62 \pm 0.08$; $P = 0.02$) (Fig. 9f, g). Fold changes also displayed a reduced trend normalized to *Rps29* (*Parp1*; control: $1.15 \pm 0.26$; Gen_direct-20: $0.60 \pm 0.10$; $P = 0.08$; *Bak1*; control: $1.05 \pm 0.13$; Gen_direct-20: $0.82 \pm 0.13$; $P = 0.25$) (Suppl. Table 2). Fold change of the apoptotic factor *Bax* normalized to *Rpl13a* was unaltered (Fig. 9h; Suppl. Table 2). The corresponding numerical values are available in Supplementary Data 1.

### Systemic DES exposure caused an upward trend in overall contraction of the mesenteric arteries

As ED is linked to cardiovascular disease, we also assessed whether systemic DES exposure could impact other vascular beds as well as the CC, therefore we analysed contraction curves of the DES_water mesenteric arteries. Control and DES_water mesenteric arteries contracted to PE and U46619 (Fig. 10a, c). Overall contraction to PE trended upwards (AUC; control: $93.83 \pm 10.26$; DES_water: $122.1 \pm 10.36$; $P = 0.08$) (Fig. 10b). PE sensitivity and maximal contraction were unaltered (Suppl. Table 1). Overall contraction to U46619

**Fig. 2 | ACh and SNP-mediated relaxation of the DES_water CC. a** Both DES_water (blue) and control (red) CC samples relaxed to acetylcholine (ACh) (1st phase). Relaxation measured as percentage of phenylephrine (PE)-induced contraction. At higher doses of ACh, DES_water and control CC samples exhibited a subsequent contraction, evident by decreasing percentage of relaxation (2nd phase). **b** Violin plot depicting significantly reduced percentage of relaxation to ACh ($10^{-5}$ M) in the DES_water CC (i.e. greater contraction) (% of PE-induced contraction; DES_water: $33.3 \pm 5.93$, $n = 10$; control: $49.3 \pm 4.73$, $n = 11$; $P = 0.04$; unpaired, two-tailed t-test). **c** Both DES_water and control CC samples relaxed to sodium nitroprusside (SNP). Data expressed as mean $\pm$ standard error of the mean (SEM), represented by error bars in (**a**) and (**c**).

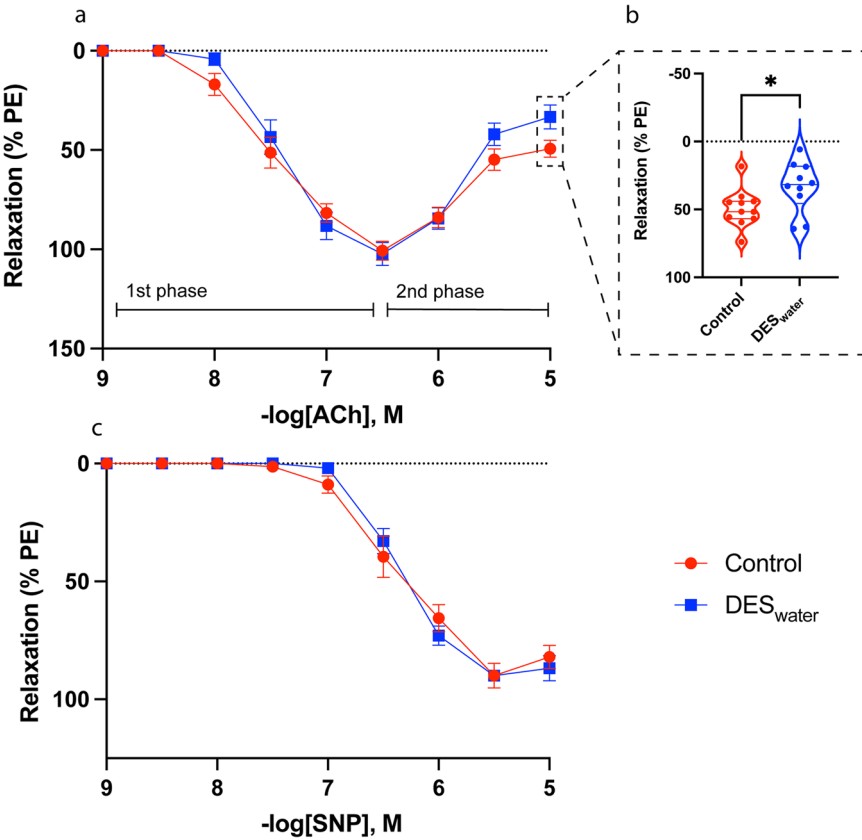

in the DES_water mesenteric arteries was also increased but statistically unaltered (control: $263.5 \pm 12.09$; DES_water: $295.5 \pm 9.404$; $P = 0.19$) (Fig. 10d). U46619 sensitivity and maximal contraction were unaltered (Suppl. Table 1). The corresponding numerical values are available in Supplementary Data 1.

## Discussion

Erectile dysfunction (ED) has a complex aetiology with many contributing environmental and lifestyle factors, the prevalence of which has been increasing globally over recent decades. Given the importance of hormonal signalling in regulating the development and function of the corpus cavernosum (CC) (a vascular structure within the penis which is critical for erection physiology), we determined whether estrogenic-EDC exposure can be considered a risk factor for ED. The potent ER agonist diethylstilbestrol (DES) was administered to mice via drinking water (DES_water) throughout development and adulthood to test the impacts of estrogenic-EDCs on erectile function. Defining the exact mechanisms of estrogenic-EDCs is complex as they can act as anti-androgens to impact erection physiology and CC patterning via disrupted testosterone output. In addition, they may also stimulate estrogen signalling directly within the CC. Thus, to control for indirect systemic effects, we also performed direct acute exposures of the CC to DES and also the phytoestrogen genistein, which is predominantly found in soy-based foods. It is important to note that the DES and genistein exposures in the present study are higher than typical environmental estrogen levels and do not necessarily reflect historic/current human exposures[89]. Rather, they represent high-dose exposures to model the widespread, constant exposures of humans to various estrogenic-EDCs. Furthermore, as this is an under-researched field it is difficult to investigate a potential causal relationship between estrogenic-EDC exposure and ED, including the underlying mechanisms, using lower doses. Thus, future studies should consider building on the findings here with other estrogenic-EDCs at different doses.

Our study demonstrated that the mouse CC elicits a biphasic response to the endothelium-dependent neurotransmitter acetylcholine (ACh) ex vivo, described here as the initial relaxation phase (1st phase) followed by a contraction phase (2nd phase); consistent with the multiphasic responses for that of the monkey and human CC[37,90]. The DES_water CC samples contracted to a greater degree in the 2nd phase of the ACh response, evident by a significantly reduced response at an ACh dose of 10 µM. These effects are endothelium-dependent as there were no differences in the response of the DES_water CC to sodium nitroprusside (SNP), an endothelium-independent NO donor. In cholinergic neurons, estrogen can modulate the expression and binding of muscarinic and potentially nicotinic ACh receptors (mAChR and nAChRs, respectively)[91,92], both of which influence CC contractility in response to ACh[93–96]. Thus, atypical density and function of these receptors within the CC may explain how DES treatment can enhance the 2nd phase of the ACh response. Overall, the phenotype demonstrates that DES treatment can augment CC contraction.

To further investigate the role of estrogenic-EDC exposure in CC contraction, we assessed the response curve of U46619, a contraction factor which mimics Thromboxane $A_2$ (TxA_2). The DES_water CC was more sensitised to this factor, suggesting that the physiological effects of TxA_2 are enhanced following systemic exposure to DES. To better understand the effects of systemic estrogenic-EDC exposure on TxA_2 signalling in the CC, we analysed expression of *Tbxar2*, the gene encoding its receptor, and found it was unaltered. Therefore, systemic DES exposure may regulate TxA_2 signalling at the non-genomic level or act downstream of this factor to enhance contraction. Given that a balance of smooth muscle relaxation and contraction pathways control CC physiology[97], estrogenic-EDC exposure may push the balance towards contraction, thereby decreasing an individual's likelihood of achieving erection and predisposing them to ED. Reports of a link between systemic estrogenic-EDC exposure and ED in humans are already emerging. Occupational exposure and urinary levels of Bisphenol A (BPA), a low potency estrogenic-EDC used in plastic

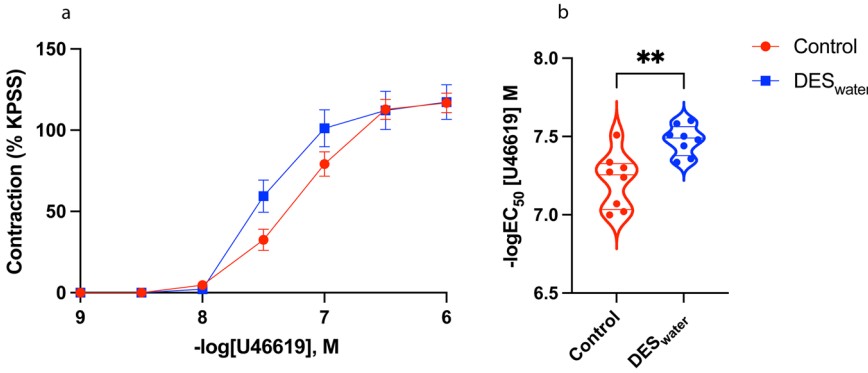

**Fig. 3 | U46619-mediated contraction response of the DES_water CC. a** U46619-mediated contraction of the DES_water CC (blue) and control CC (red). Contraction measured as percentage of high potassium physiological saline solution (KPSS)-induced contraction. **b** Violin plot depicting significantly greater sensitivity of the DES_water CC to U46619 ($-\log$EC$_{50}$ [U46619] M; DES_water: $7.48 \pm 0.03$, $n = 8$; control: $7.22 \pm 0.06$, $n = 8$; $P = 0.003$; unpaired, two-tailed t-test). Data expressed as mean $\pm$ standard error of the mean (SEM), represented by error bars in (**a**). Significance ($P < 0.05$) indicated by **.

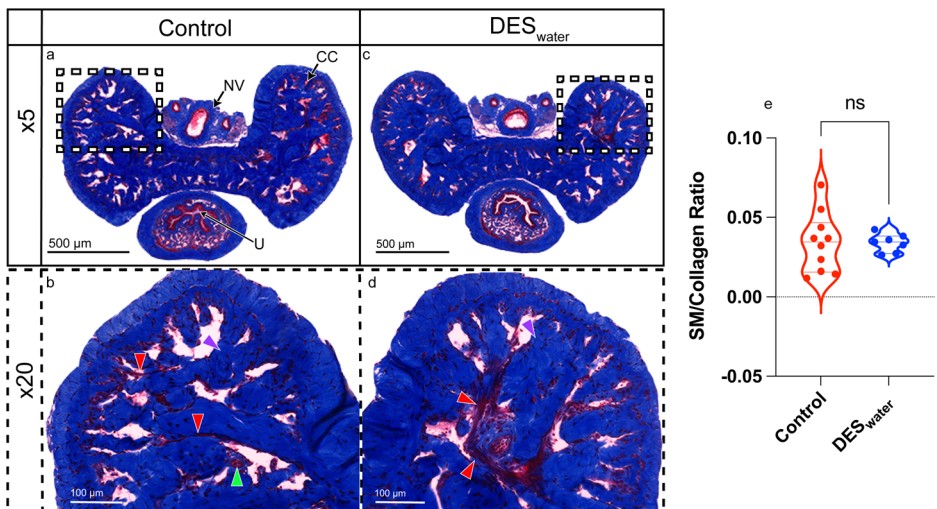

**Fig. 4 | Histological analysis demonstrating no difference in smooth muscle to collagen ratio of DES_water CC.** Sections stained with Masson's Trichrome and imaged with SlideViewer 2.6. **a** ×5 magnification of control penis shaft indicating the dorsal neurovascular bundle (NV), the corpus cavernosum (CC) and urethra (U). Dotted box indicates portion of image magnified for panel (**b**). **b** ×20 magnification of the control CC. Red arrows indicate vasculature or smooth muscle. Green arrow indicates the cavernous artery. Purple arrow indicates collagen. **c** ×5 magnification of DES_water penis shaft. Dotted box indicates portion of image magnified for panel (**d**). **d** ×20 magnification of the DES_water CC. **e** Violin plot depicting no significant difference for smooth muscle (SM) to collagen ratio in DES_water CC histological samples (blue) (smooth muscle area %/collagen area %; control (red): $0.03 \pm 0.01$, $n = 10$; DES_water: $0.03 \pm 0.002$, $n = 7$; $P = 0.97$; unpaired, two-tailed t-test). Data expressed as mean $\pm$ standard error of the mean (SEM). ns: not significant.

production, has been associated with increased likelihood of ED[98,99]. These studies speculated that BPA may function as an anti-androgen or induce epigenetic effects to interfere with male sexual function, although provided little comment on potential estrogenic effects directly in the erectile tissue.

Our study included assays for testosterone output to distinguish potential anti-androgenic effects on contraction from true estrogenic effects on the CC. However, we observed no differences in the AGD, testes weight, seminal vesicle weight, CC smooth muscle to collagen ratio, androgen receptor (*Ar*) expression in the CC or plasma testosterone concentration; all of which are key markers for androgen production. These results demonstrate that androgen production was not disrupted in the DES_water mice. In contrast, rat pups exposed to DES exhibit a decline in circulating testosterone and a reduction in CC smooth muscle cell proliferation[100]. However, differences in the drug dose and administration method may explain the contradictory findings.

Conversely, *Esr1* mRNA (encoding ERα) was significantly increased within the DES_water CC, consistent with the increased transcription and protein levels of *Esr1* in the rat penis and CC following developmental DES exposure[70,100]. These findings are consistent with the role of ERα in mediating DES-induced structural abnormalities of the rat CC[69], as well as the responsiveness of the human ERα gene (*ESR1*) to estrogen[101]. The results are not entirely consistent with other tissue types. For example, the mouse

embryonic gonads and the rat hypothalamus exhibit no changes in transcriptional activity of *Esr1* following developmental exposure to DES[102,103]. Furthermore, although DES can increase *Esr1* mRNA levels in the uterine epithelium of neonatal mice, it has no such effect on the uterine mesenchyme[104]. Thus, estrogenic-EDCs such as DES may exert tissue-specific effects on the CC. Given that the blockade of estrogen signalling can reduced the magnitude of CC contraction in the rabbit[54], it is logical that increased protein levels of ERα augment contraction of the DES_water CC. However, future studies should verify this by assessing ERα protein levels in the mouse CC following DES exposure.

The molecular mechanism between increased local expression of *Esr1* and amplified CC contraction require further investigation as we observed no significant differences for the expression of *Ar* (discussed above), *Tbxar2* (discussed above), *Nos3, Angpt4, Sfrp1, Igfbp3,* and *Oxtr1*; genes which are implicated in erection physiology, ED and/or penis development[54,105–109]. Of these, estrogen signalling is reported to transcriptionally regulate *Ar, Tbxar2, Nos3, Sfrp1,* and *Igfbp3* in other tissue types[64,110–113], although our study suggests they are not impacted by systemic estrogenic-EDC exposure in the CC. Conversely, it is unclear whether estrogen transcriptionally regulates *Angpt4* and *Oxtr1*[114,115]. These genes were included in the present study to ascertain any broader or indirect effects that systemic DES exposure may have on CC function. Overall, our data demonstrates that a systemic

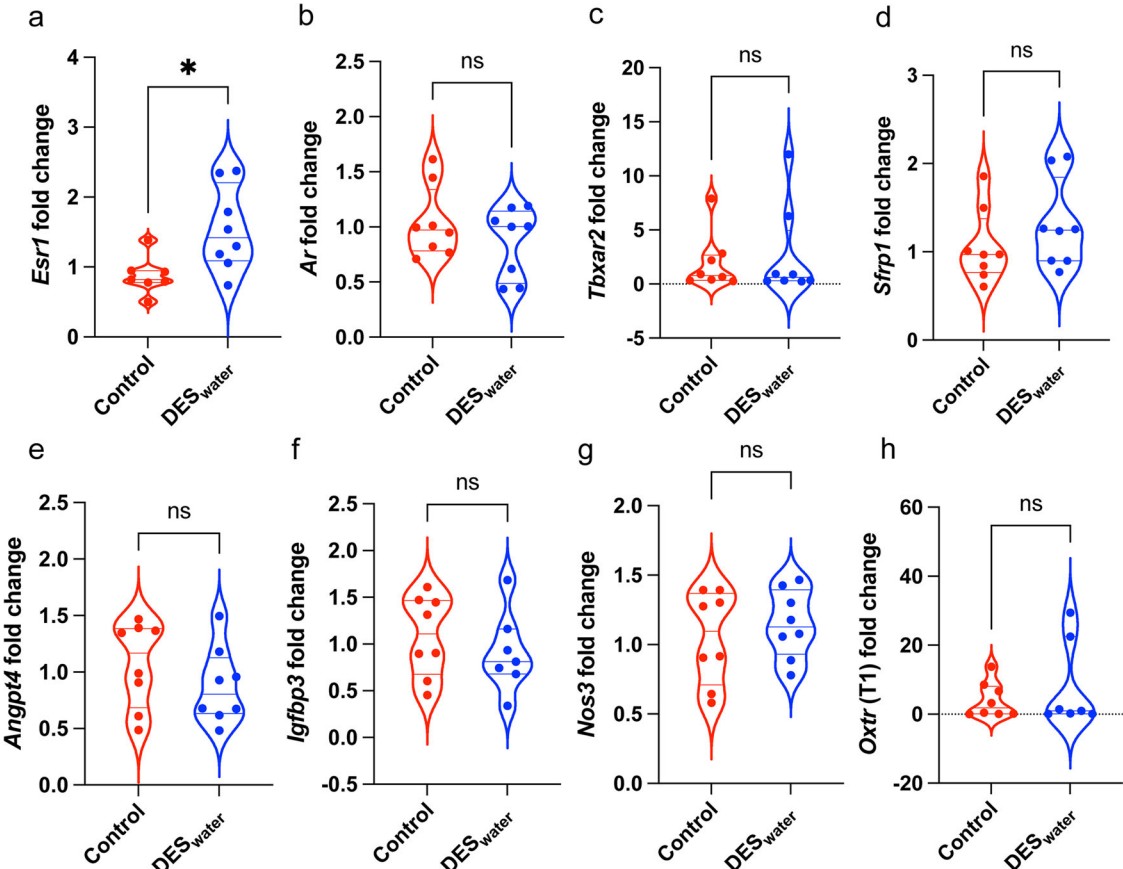

**Fig. 5 | RT-PCR results from the DES_water CC. a–h** Violin plots of gene expression fold change of the control (red) and DES_water CC normalized to the housekeeper *Rpl13a* in RT-PCR. **a** *Esr1* is significantly upregulated in the DES_water CC (control: $0.88 \pm 0.10$, $n = 7$; DES_water: $1.54 \pm 0.21$, $n = 8$; $P = 0.02$; unpaired, two-tailed t-test). **b** *Ar* (control: $1.04 \pm 0.11$, $n = 8$; DES_water: $0.87 \pm 0.11$, $n = 8$; $P = 0.72$; Mann–Whitney test). **c** *Tbxar2* (control: $1.94 \pm 0.92$, $n = 8$; DES_water: $2.66 \pm 1.52$, $n = 8$; $P = 0.72$; Mann–Whitney test). **d** *Sfrp1* (control: $1.06 \pm 0.15$, $n = 8$; DES_water: $1.31 \pm 0.18$, $n = 8$; $P = 0.33$; Mann–Whitney test). **e** *Angpt4* (control: $1.07 \pm 0.13$,

$n = 8$; DES_water: $0.88 \pm 0.12$, $n = 8$; $P = 0.30$; unpaired, two-tailed t-test). **f** *Igfbp3*: (control: $1.09 \pm 0.15$, $n = 8$; DES_water: $0.91 \pm 0.16$, $n = 7$; $P = 0.43$; unpaired, two-tailed t-test). **g** *Nos3* (control: $1.05 \pm 0.12$, $n = 8$; DES_water: $1.15 \pm 0.09$, $n = 8$; $P = 0.52$; unpaired, two-tailed t-test). **h** *Oxtr* transcript 1 (T1) (control: $4.13 \pm 1.81$, $n = 8$; DES_water: $7.82 \pm 4.75$, $n = 7$; $P = 0.78$; Mann–Whitney test). Data expressed as mean ± standard error of the mean (SEM). Significance ($P < 0.05$) indicated by *, ns = non-significant.

estrogenic-EDC exposure does not necessarily affect androgen production yet can enhance endogenous estrogen signalling localised to the CC, which likely causes increased contraction. Future studies should investigate the molecular signalling pathway which underlies this estrogen-mediated CC contraction. Our findings align with reports that testosterone therapy is not always effective in treating ED[85], suggesting a direct mechanism by which estrogenic-EDCs may contribute to ED.

We next investigated the direct effects of DES exposure on CC function, as well as that of the phytoestrogen genistein, which is primarily found in soy products and is a less potent estrogen than DES. Direct DES exposure at 10 µM (DES_direct-10) caused complete dysfunction of the CC. After reducing the dose to 5 µM (DES_direct-5), we found a significant desensitization to ACh, demonstrating an endothelium-dependent deleterious effect on CC physiology. As already discussed in the context of the DES_water CC, the effects of estrogen on elements of cholinergic signalling within the CC (such as the mAChRs and the nAChRs) may contribute to this phenotype. Surprisingly however, the maximal relaxation response to SNP was significantly increased in the DES_direct-5 CC, suggesting that DES exposure directly enhances relaxation of the CC independently of endothelium. However, the result may simply represent the inability of DES_direct-5 CC samples to maintain tension[116], as was observed in the higher dose (DES_direct-10 CC). Either way, these data demonstrate that a direct exposure to an estrogenic-EDC can cause CC dysfunction.

Similarly, samples exposed directly to the weaker estrogenic-EDC genistein (Gen_direct-20) showed a reduced 2nd phase of the ACh response, although not ablated as was observed for the DES_direct-5 CC. These effects were accompanied by significant desensitization to ACh, but not SNP, suggesting that DES and genistein target the endothelium. The reduced trend for maximal relaxation in response to ACh following genistein exposure supports this notion. Furthermore, the Gen_direct-20 CC exhibited significantly reduced *Nos3* expression, a gene which encodes endothelial NO synthase (eNOS), a principal factor in mediating endothelial-dependent CC relaxation. For instance, genetic knockout of *Nos3* in mice not only diminishes the ACh response in the CC ex vivo, but causes it to contract when it would otherwise relax[117]. Therefore, direct DES and genistein exposure at these doses may induce endothelial dysfunction, a condition described as an imbalance of endothelium-derived contraction and relaxation factors, in particular NO[118]. Endothelial dysfunction is a condition with established links to ED as well as cardiovascular disease[118,119], further implicating direct estrogenic-EDC exposures as contributing factors. In addition, high consumption of isoflavones (phytoestrogens) in soy products (310–361 mg/day for 1–3 years) is reported to cause ED and may result from the decreased testosterone levels also reported in these individuals[79–81]. However, our study provides an additional possible mechanism for these results, whereby genistein can interfere with CC function directly.

**Fig. 6 | ACh- and SNP-mediated relaxation of the DES$_{direct-5}$ CC. a** Acetylcholine (ACh)-mediated relaxation curve of DES$_{direct-5}$ CC (blue) and control (red) samples. Relaxation responses measured as percentage of phenylephrine (PE)-induced contraction. Unlike the control CC, the DES$_{direct-5}$ CC exhibited only the 1$^{st}$ phase (relaxation) and not the 2$^{nd}$ phase (contraction) of the ACh response. **b** Violin plot depicting significantly reduced sensitivity of the DES$_{direct-5}$ CC to ACh (DES$_{direct-5}$: 6.61 ± 0.08, $n = 9$; control: 7.31 ± 0.06, $n = 9$; $P = 0.0001$; paired, two-tailed t-test). **c** Sodium nitroprusside (SNP)-mediated relaxation curve of the DES$_{direct-5}$ CC. **d** Violin plot depicting significantly higher maximal relaxation to SNP in the DES$_{direct-5}$ CC (DES$_{direct-5}$: 93.15 ± 5.36, $n = 10$; control: 72.88 ± 7.23, $n = 10$; $P = 0.04$; Wilcoxon matched pairs test). Data expressed as mean ± standard error of the mean (SEM).

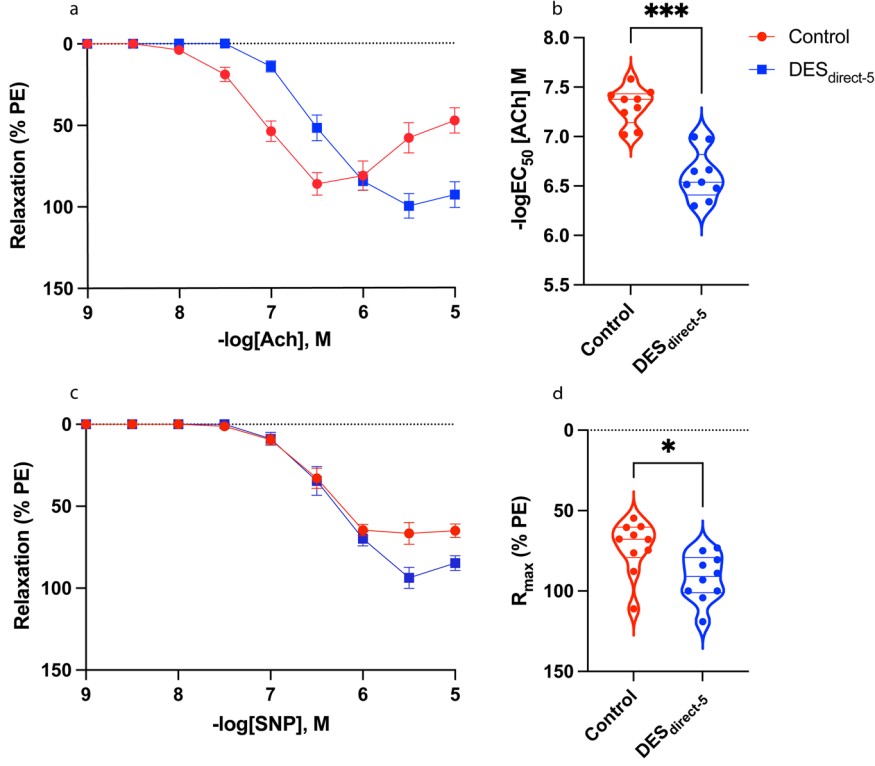

To investigate the mechanisms underlying the DES$_{direct-5}$ and Gen$_{direct-20}$ CC phenotypes, we analysed expression of *Ar*, *Esr1*, and *Nos3*. The anti-apoptotic markers *Bcl-XL* and *Bcl2*, as well as the apoptotic markers *Parp1*, *Bax*, and *Bak1* were also included to assess cytotoxic affects following acute exposures. Of these markers, estrogen signalling is reported to transcriptionally regulate *Bcl-XL*, *Bcl2* and *Bak1*[120–122]. It is unclear whether this is also true for *Parp1* or *Bax*[123,124], although these were included to screen for any indirect cytotoxic effects of estrogenic-EDCs. In the DES$_{direct-5}$ CC, none of these factors exhibited significant differences in expression. The small sample size potentially explains such results, although the duration of DES exposure was likely too short to elicit a genomic response. Instead, its actions may have triggered non-genomic estrogenic responses as reported in acute exposures in the gonads[125,126]. Supporting an estrogenic mechanism is the comparatively muted reduction in sensitivity to ACh of the Gen$_{direct-20}$ CC relative to the DES$_{diret-5}$ CC, which is consistent with the relative estrogenic potencies of these EDCs[127]. It is also important to note that genistein is a potent flavonoid which can elicit estrogen-independent effects, such as its role as a tyrosine kinase inhibitor and phosphodiesterase inhibitor[82,83]. However, inhibition of these factors is associated with improved erectile function in rodents[128–131], directly contrasting with impaired relaxation of the Gen$_{direct-20}$ CC and reduced *Nos3* expression. Thus, the phenotype reported in the present study is more likely to reflect the estrogenic action of genistein.

The Gen$_{direct-20}$ CC exhibited no differences in expression for *Esr1* and *Ar*, although it was significantly reduced for *Nos3* as described above. In addition, there were significant reductions in expression of *Bcl-XL*, *Bcl2*, *Parp1*, and *Bak1* (but not *Bax*). Thus, the high dose of genistein may have stimulated cytotoxic pathways dependently or independently of classic ER signalling, as described above. This potentially explains the reduced *Nos3* expression and the endothelium-dependent abnormal CC contractility. Similarly, the estrogenic-EDC BPA is reported to elicit oxidative stress associated with DNA damage of rat insuloma cells[132]. Although the direct DES and genistein treatment doses were higher than what is commonly found in human serum levels[133,134], it is important for understanding their mechanism of action. Our data demonstrate that high doses of DES and genistein can directly and severely disrupt CC function.

The results from the systemic DES exposure study contrasted with that of the direct estrogenic-EDC exposures. Unlike the DES$_{direct-5}$ and Gen$_{direct-20}$ treatment groups, the DES$_{water}$ CC did not exhibit reduced sensitivity to endothelial-dependent relaxation factors. In addition, the DES$_{water}$ CC exhibited increased *Esr1* expression with no alteration to *Nos3*; the opposite of what was observed in the Gen$_{direct-20}$ CC which also exhibited markers for cytotoxicity. These findings align with prior research demonstrating that estrogenic-EDCs at lower dose ranges can increase estrogen signalling, whereas higher doses induce cytotoxicity[135], which also relates to the non-monotonic effects characteristic of several estrogenic-EDCs[136–141]. For example, in utero exposures to low doses of genistein are associated with DSDs such as hypospadias in mice and humans[142–145], demonstrating that even very low doses of estrogenic-EDCs can impact development. In the context of ED, our study reiterates the importance of testing a range of doses and treatment methods of EDCs to understand their full spectrum of effects.

ED is identified as an independent risk factor for cardiovascular disease, therefore mesenteric arteries were also analysed in this study to determine if the effects of dietary estrogenic-EDC exposure on the CC would mirror that of other vascular beds[146]. The responses of DES$_{water}$ mesenteric arteries to U46619 and PE were statistically unaltered. However, we observed a trend for increased overall contraction to PE, with the same direction observed for that of the U46619 response. The trends align with reports of increased frequency of cardiovascular-related mortality in prostate cancer patients receiving daily oral DES supplements[147,148]. In addition, out of 30 patients evaluated for toxicity of DES treatment, mesenteric artery thrombosis was reported as a cause of death in one individual with no reported history of cardiovascular disease; potentially attributed to the treatment[149]. In our study, the CC may be more sensitive to environmental insult compared to more robust blood vessels[150,151], possibly explaining the lack of significance in the increased contraction of the DES$_{water}$ mesenteric arteries. Indeed, a recent single-cell RNA sequencing study demonstrated that a subpopulation of endothelial cells in the human CC are more susceptible to injury compared to adjacent subpopulations and occur at higher

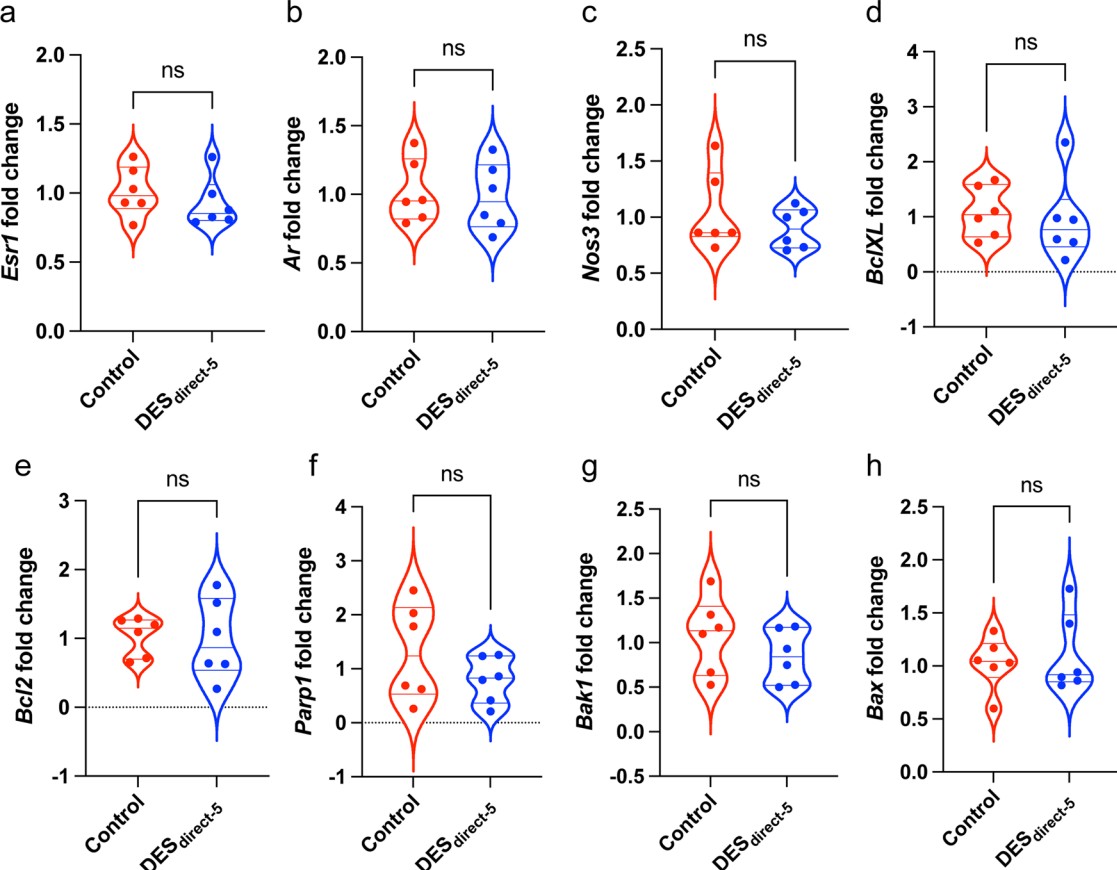

**Fig. 7 | RT-PCR results from the DES$_{direct-5}$ CC. a–h** Violin plots of gene expression fold change of the control (red) and DES$_{direct-5}$ CC (blue) normalized to the housekeeper *Rpl13a* in RT-PCR. All results are not significant. **a** *Esr1* (control: 1.01 ± 0.07, n = 6; DES$_{direct-5}$, n = 6: 0.93 ± 0.07; P = 0.42; unpaired, two-tailed t-test). **b** *Ar* (control: 1.02 ± 0.09, n = 6; DES$_{direct-5}$: 1.00 ± 0.10, n = 6; P = 0.77; unpaired, two-tailed t-test). **c** *Nos3* (control: 1.04 ± 0.14, n = 6; DES$_{direct-5}$: 0.90 ± 0.07, n = 6; P = 0.59; Mann–Whitney test). **d** *BclXL* (control: 1.09 ± 0.19, n = 6; DES$_{direct-5}$:

0.94 ± 0.31, n = 6; P = 0.69; unpaired, two-tailed t-test). **e** *Bcl2* (control: 1.04 ± 0.11, n = 6; DES$_{direct-5}$: 0.99 ± 0.24, n = 6; P = 0.86; unpaired, two-tailed t-test). **f** *Parp1* (control: 1.31 ± 0.37, n = 6; DES$_{direct-5}$: 0.79 ± 0.17, n = 6; P = 0.23; unpaired, two-tailed t-test). **g** *Bak1* (control: 1.08 ± 0.17, n = 6; DES$_{direct-5}$: 0.84 ± 0.12, n = 6; P = 0.30; unpaired, two-tailed t-test). **h** *Bax* (control: 1.03 ± 0.10, n = 6; DES$_{direct-5}$: 1.11 ± 0.15, n = 6; P = 0.82; Mann–Whitney test). ns = non-significant.

proportions in ED patients[152]. Therefore, the balance of contraction and relaxation factors in the CC may initially have been more susceptible to systemic DES exposure than that of the mesenteric arteries. Overall, the predisposition to ED that we observed following systemic DES exposure may extend into the wider vascular system.

It is also important to note that this study analysed CC physiology of mice in the young adult stage which equate to ~20–30 years of age for humans[153]. Given that major cardiovascular events typically occur 3–5 years after the initial onset of ED and at older ages, the DES$_{water}$ mice may have experienced an initial predisposition to ED without yet developing other forms of cardiovascular disease[4,154,155]. Our results stress a need for epidemiological studies to incorporate ED of physiological and organic origins in younger men, rather than of older men alone to which this form of the condition is normally ascribed[156].

Estrogenic-EDC exposures via diet significantly increased CC contraction in young, otherwise normal adult male mice, suggesting that chronic estrogenic-EDC exposure predisposes individuals to ED. These results were associated with normal masculinisation (i.e., unaffected androgen production) but increased expression levels of the gene encoding ERα within the CC, suggesting that systemic estrogenic-EDC exposure can impact CC physiology directly. To investigate further, we developed a method of incubating isolated wild-type CC samples in DES and genistein which primarily led to endothelial dysfunction of the CC. Thus, acute exposure to estrogenic-EDCs is likely to directly increase ED risk in previously unexposed individuals. Although both systemic and direct

estrogenic-EDC exposures impacted CC function and gene expression, the effects were not uniform, which likely reflects the different doses and durations of these treatment regimes. Finally, we uncovered a potential link between estrogenic-EDC exposure and system-wide vascular dysfunction. The present findings contribute to a sparse group of studies which demonstrate that estrogenic-EDCs can impact CC contractility. Future studies should continue this line of research with different estrogenic-EDCs, administration methods and doses which reflect environmental exposures. Taken together, EDCs should be considered and further investigated as an additional independent risk factor for ED, potentially extending into cardiovascular disease.

## Methods
### Animals, tissue collection and EDC treatments
Wild-type C57BL/6 mice (Jackson Laboratory) were housed in the Biosciences 4 animal facility at the University of Melbourne. Mice were maintained on a 12 h light/dark cycle at 20 °C, with standard food pellets (Barastock, VIC, Australia) and water available *ad libitum*. Previous studies have demonstrated that the C57BL/6 background is suitable for modelling ED[157–159] and developmental estrogenic endocrine disruption in humans[160]. Mice were handled and killed according to the Australian code for the care and use of animals for scientific purposes established by the National Health and Medical Research Council (2013). All protocols and experiments were approved by the University of Melbourne Animal Ethics Committee. We have complied with all relevant regulations for animal use.

**Fig. 8 | ACh- and SNP-mediated relaxation of the Gen$_{direct-20}$ CC. a** Acetylcholine (ACh)-mediated relaxation response of Gen$_{direct-20}$ (green) and control (red) CC samples. Relaxation measured as percentage of phenylephrine (PE)-induced contraction. Both the control and Gen$_{direct-20}$ CC displayed both phases of the ACh response, although the 2$^{nd}$ phase (contraction) was less pronounced in the Gen$_{direct-20}$ CC. **b** Violin plot depicting significantly reduced sensitivity of the Gen$_{direct-20}$ CC to ACh ($-$logEC$_{50}$ [Ach] M; Gen$_{direct-20}$: 7.34 ± 0.09, $n = 7$; control: 7.64 ± 0.07, $n = 7$; $P = 0.002$; unpaired, two-tailed t-test). **c** Sodium nitroprusside (SNP)-mediated relaxation curve of the Gen$_{direct-20}$ and control CC. Data expressed as mean ± standard error of the mean (SEM), represented by error bars in (**a**) and (**c**). Significance ($P < 0.05$) indicated by **.

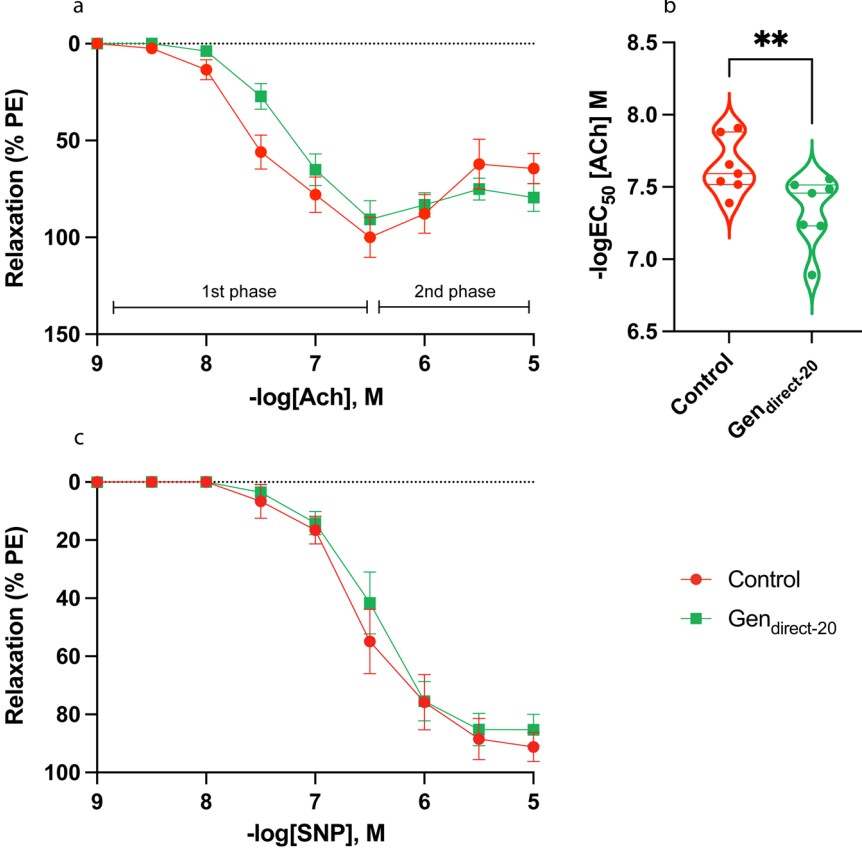

Diethylstilbestrol (DES) (Abcam) was administered in mouse drinking water (2 µg/mL) and replaced every 2 days. Although this represents a dose greater than what is reported for concentrations of individual estrogenic-EDCs in the environment[161,162] and does not necessarily reflect historical or current DES exposures in humans, it was used to first define the pathways by which estrogenic-EDCs can potentially contribute to ED, whether they be via direct ER activation within the CC or reduced testosterone output from the testes. Timed pregnant mice were exposed to DES in drinking water starting at embryonic day (E) 9.5. DES treatment continued in the pups (DES$_{water}$ group: $n = 39$) (where exposure occurred via breast milk of the mother), after weaning and into adulthood when the males were humanely killed. The temporal window of treatment was to ensure that exposure occurred throughout sex determination, early testes development and CC development. In pregnant rodents, DES can cross the placental barrier and compared to estradiol, exhibits reduced metabolism and greater accumulation in fetal tissue, thus it is appropriate for modelling embryonic exposures to estrogenic-EDCs[163,164]. A vehicle control was also examined where equal amount of the DES diluent (100% ethanol) was added into drinking water and refreshed every 2 days (control: $n = 41$). DES$_{water}$ mice and the corresponding negative controls were humanely killed by cervical dislocation between P64-96.

Several assays were used to test whether DES exposure reduced testosterone production from the testes. Blood samples were collected via cardiac puncture using a 26 G needle. Plasma was separated by centrifugation at 3000 rpm for 10 min at 4 °C. Plasma samples were sent to the Anzac Research Institute (Andrology Department) and testosterone concentration was measured by steroid LC-MS (DES$_{water}$: $n = 26$; control: $n = 27$). Anogenital distance (AGD) (mm; DES$_{water}$: $n = 39$; control: $n = 39$), testes weight (g; DES$_{water}$: $n = 39$; control: $n = 39$) and seminal vesicle weight (g; DES$_{water}$: $n = 41$; control: $n = 39$) were recorded.

### Acute DES and genistein exposures

For the acute direct EDC exposures, CCs from untreated C57BL/6 wild-type male mice at P67-77 (~9–11 weeks of age) were incubated in an estrogenic-EDC or vehicle solution. As the CC is a paired structure, one CC from each mouse was allocated to a treatment group and the other to the corresponding vehicle control group, allowing for pairwise statistical analyses.

DES was dissolved in 100% EtOH to make a stock concentration (0.1 M). This was diluted first in 50% ethanol then serial diluted in DMEM supplemented with penicillin/streptomycin (1X) to working concentrations (5 µM and 10 µM); CC samples exposed to these treatments are referred to as the DES$_{direct-5}$ ($n = 16$) and DES$_{direct-10}$ ($n = 8$) groups, respectively.

Genistein (Abcam) was dissolved in 100% DMSO at a stock concentration (0.1 M). This was diluted first in 50% DMSO then serial diluted DMEM supplemented with penicillin/streptomycin (1X) to a working concentration (20 µM). CC samples exposed to this medium are referred to as the Gen$_{direct-20}$ group ($n = 13$). For the vehicle-treated controls, drug stock solution was substituted for an equal amount of 100% ethanol or DMSO according to the corresponding treatment (DES$_{direct-5}$ control $n = 16$; DES$_{direct-10}$ control = 8; Gen$_{direct-20}$ control $n = 13$).

The direct estrogenic-EDC doses are greater compared to typical environmental exposures[89] and were implemented to investigate potential pathways by which estrogenic-EDCs may contribute to ED.

Isolated CC samples were individually suspended in 1 mL of treatment or vehicle working solution in a 24-well plate, incubated at 37 °C and gassed with 5% $CO_2$ and atmospheric $O_2$ for 4 h. For CC samples used for RT-PCR analysis, initial incubation in growth media was followed by incubation in the same conditions but in 5 mL Krebs buffer (see below) for an additional 3 h before being snap-frozen. This was to replicate the conditions corresponding to functional testing with wire myography.

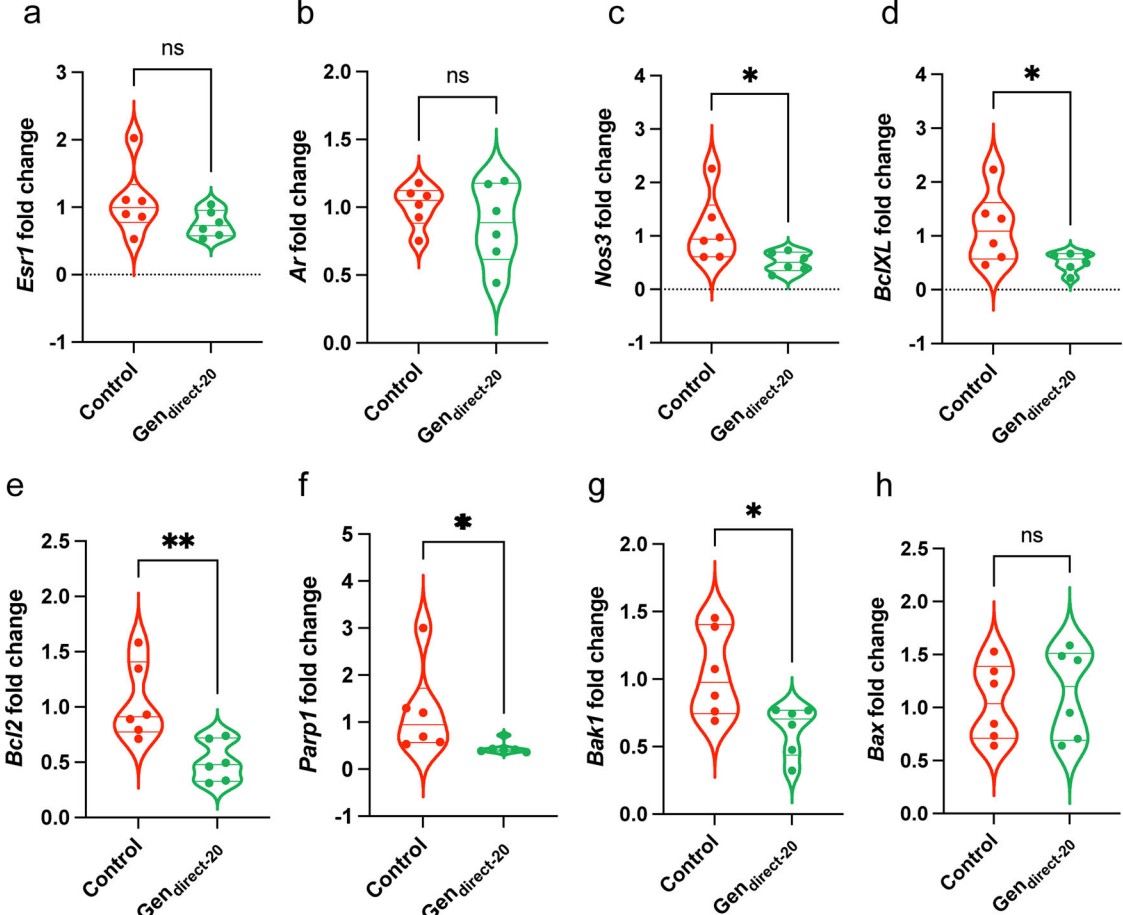

**Fig. 9 | RT-PCR results from the Gen$_{direct-20}$ CC. a–h** Violin plots of gene expression fold change of the control (red) and Gen$_{direct-20}$ CC (green) normalized to the housekeeper *Rpl13a* in RT-PCR. **a** *Esr1* (control: 1.09 ± 0.21, $n = 6$; Gen$_{direct-20}$: 0.76 ± 0.08, $n = 6$; $P = 0.17$; unpaired, two-tailed t-test). **b** *Ar* (control: 1.01 ± 0.06, $n = 6$; Gen$_{direct-20}$: 0.88 ± 0.12, $n = 6$; $P = 0.34$; unpaired, two-tailed t-test). **c** *Nos3* (control: 1.12 ± 0.25, $n = 6$; Gen$_{direct-20}$: 0.51 ± 0.08, $n = 6$; $P = 0.046$; unpaired, two-tailed t-test). **d** *BclXL* (control: 1.15 ± 0.27, $n = 6$; Gen$_{direct-20}$: 0.52 ± 0.07, $n = 6$; $P = 0.046$; unpaired, two-tailed t-test). **e** *Bcl2* (control: 1.04 ± 0.14, $n = 6$; Gen$_{direct-20}$:

0.51 ± 0.07, $n = 6$; $P = 0.01$; unpaired, two-tailed t-test). **f** *Parp1* (control: 1.22 ± 0.38, $n = 6$; Gen$_{direct-20}$: 0.46 ± 0.13, $n = 6$; $P = 0.02$; Mann–Whitney test). **g** *Bak1* (control: 1.04 ± 0.13, $n = 6$; Gen$_{direct-20}$: 0.62 ± 0.08, $n = 6$; $P = 0.02$; unpaired, two-tailed t-test). **h** *Bax* (control: 1.05 ± 0.15, $n = 6$; Gen$_{direct-20}$: 1.14 ± 0.17, $n = 6$; $P = 0.72$; unpaired, two-tailed t-test). Gene expression data normalized to the housekeeper *Rpl13a*. Unpaired, two-tailed t-tests used for statistical analysis. Data expressed as mean ± standard error of the mean (SEM). Significance ($P < 0.05$) indicated by *,**. ns = non-significant.

## Wire myography-CC

Following cervical dislocation, the glans penis was removed, and the penis shaft dissected free. The CC was removed by dissecting the neurovascular bundle and urethra from the shaft of the penis. A sagittal cut through the midline separated the paired CC. One CC was isolated in ice-cold Krebs physiological salt solution (PSS) containing (mM): NaCl (120), NaHCO$_3$ (25), KCl (5), MgSO$_4$ (1.2), KH$_2$PO$_4$ (1), D-glucose (11.1), and CaCl$_2$ (2.5) (pH 7.4) for use in wire myography. The paired CC was snap-frozen and stored at −80 °C for gene expression analyses.

Wire myography establishment protocol was optimized for the mouse CC: each CC sample was placed in a myograph channel organ bath and immersed in ice-cold Krebs buffer. The samples were then mounted in the longitudinal axis to a four-channel wire myograph (Danish Myo Technology, Aarhus, Denmark) by puncturing both ends of the CC with the myograph channel pins which were set at 2 mm from each other. The tissue was then washed with warmed Krebs at 37 °C bubbled with carbogen. The tissue was equilibrated for 30 min, replacing Krebs buffer every 10 min. The tissue was then gradually stretched to a preload tension of 4.2 mN (0.45 g) in a stepwise manner. The tissue was then left for 60 min, replacing the Krebs buffer at 20 min. intervals. CCs were then tested with either the relaxation or contraction assays.

**CC relaxation assay.** Phenylephrine (PE) (5 μM) was added to activate the tissue before washing out, to test tissue viability. Tissue relaxation was then checked by contraction with PE (5 μM) again then adding acetylcholine (ACh) (1 μM) to induce relaxation. If the tissue failed to contract and relax from these tests, it was considered non-viable and omitted from data analysis. Viable CC samples were then subjected to a PE (5 μM) -induced contraction for each of the following ACh (0.1 nM to 10 μM) and sodium nitroprusside (SNP) (0.1 nM to 10 μM) concentration response-curves to assess endothelium-dependent and -independent smooth muscle relaxation, respectively. Relaxation responses to each dose of ACh and SNP was measured as a percentage of the corresponding PE-induced contraction ([P64-72; DES$_{water}$: $n = 11$; control: $n = 10$], [P71-77; DES$_{direct-10}$: $n = 8$; control: $n = 8$], [P67-76; DES$_{direct-5}$: $n = 10$; control: $n = 10$], [P67-75; Gen$_{direct-20}$: $n = 7$; control: $n = 7$]).

**CC contraction assay.** A high potassium physiological saline solution (KPSS, isotonic replacement of Na$^+$ with K$^+$) (120 mM) was added to induce contraction and washed out. Any samples which did not respond to KPSS were omitted from data analysis. Concentration-response curves to U46619 (0.1 nM to 1 μM) were established for the CC. Contraction responses to each dose of U46619 was measured as a percentage of KPSS-induced contraction (P78-90; DES$_{water}$: $n = 8$; control: $n = 8$).

**Fig. 10 | PE- and U46619-mediated contraction of the DES$_{water}$ mesenteric artery. a** Phenylphrine (PE)-mediated contraction curve of DES$_{water}$ (blue) and control (red) mesenteric artery. Contraction measured as percentage of high potassium physiological saline solution (KPSS)-induced contraction. **b** Violin plot demonstrating an increased trend of overall contraction to PE in the DES$_{water}$ mesenteric arteries (Area Under the Curve [AUC] of PE; control: 93.83 ± 10.26, $n = 7$; DES$_{water}$: 122.1 ± 10.36, $n = 8$; $P = 0.08$; unpaired, two-tailed t-test). **c** U46619-mediated contraction curve of the DES$_{water}$ and control mesenteric arteries. **d** Violin plot of overall contraction to U46619 demonstrating an increased trend in the DES$_{water}$ mesenteric arteries (AUC of U46619; control: 263.5 ± 12.09, $n = 7$; DES$_{water}$: 295.5 ± 9.404, $n = 8$; $P = 0.19$; Mann–Whitney test). Data expressed as mean ± standard error of the mean (SEM), represented by error bars in (**a**) and (**c**).

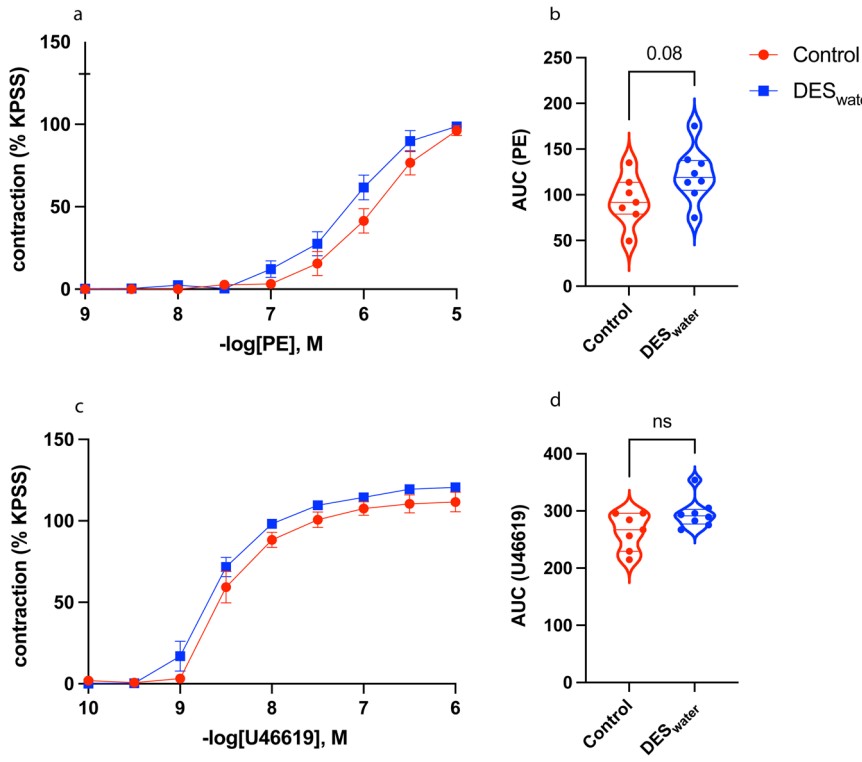

## Wire myography-mesenteric artery

In addition to the CC, contraction responses of the mesenteric arteries were also tested. The mesenteric artery was isolated and placed in ice-cold Krebs. Small mesenteric arteries (first-order branches of the superior mesenteric artery) were isolated, cleared of loose connective and adipose tissue, and cut into 2 mm rings in line with prior research[165]. The artery sections were mounted on the myograph and allowed to equilibrate for 15 min before normalization. To test tissue viability, arteries were first exposed to high 100 mM KPSS and then washed out. Subsequently, the integrity of the endothelium was determined by pre-constricting arteries with PE (0.1–3 μmol/L) to 50–70% of KPSS contraction, then applying the endothelium-dependent vasodilator ACh (10 μmol/L). Arteries with >90% relaxation were deemed suitable to continue. Concentration-response curves to PE (0.1 nM to 10 μM) and U46619 were established for the mesenteric arteries as described for that of the CC (P78-90; DES$_{water}$: $n = 8$; control: $n = 8$).

## Wire myography parameters

Samples from each wire myography drug-response curve were analysed by measuring sensitivity ($-\log EC_{50}$), overall relaxation/contraction (Area Under the Curve [AUC]), and maximal relaxation/contraction (maximum % reduction of PE-induced contraction, or % of KPSS-induced contraction [R$_{max}$/C$_{max}$]). Few studies have assessed the CC as we have in this study. Based on the limited work of others and our previous work using the technique of wire myography, an $n = 5$–8 is often sufficient to demonstrate relevant statistical differences[165–167].

## Histology

The midshaft of the penis was isolated and immediately fixed in Bouin's solution (Sigma) for 1–2 h at room temperature, before being processed in paraffin and sectioned at 7 μm. The slides were stained with Masson's trichrome (to differentiate smooth muscle from collagen) and scanned with a Pannoramic Scan II Digital Slide Scanner (3D Histech, Hungary) and viewed with SlideViewer 2.6 (3D Histech, Hungary). On the viewing software, the dorsal region of the CC was imaged at ×20 and enhanced for red colouration at a setting of 0.7 to enhance contrast between collagen and smooth muscle. Images were analysed using QuPath (0.3.2) software and set to brightfield (other) image type. Colour threshold values are indicated in Supplementary Table 3. For each biological replicate, smooth muscle to collagen ratios were calculated for the dorsal CC region in triplicate and averaged (P66-93; DES$_{water}$: $n = 7$; control: $n = 10$).

## RT-PCR

Total RNA was extracted from CC samples ([P64-90; DES$_{water}$: $n = 8$; control: $n = 8$], [P77; DES$_{direct-5}$: $n = 6$; control: $n = 6$], [P76-77, Gen$_{direct-20}$: $n = 6$; control: $n = 6$]) using the GenElute™ Mammalian Total RNA Miniprep Kit (Sigma) according to the manufacturer's instructions for fibrous tissue with the following modifications: samples were homogenized in lysis solution (300 μL) for 90 s. Nuclease-free water (590 μL) and Proteinase K solution (10 μL) (Thermo Fisher) was added to each sample and incubated at 55 °C for 30 min.

Genomic DNA was removed from RNA samples using the DNA-*free*™ DNA Removal Kit (Invitrogen) and cDNA synthesised with the Superscript™ IV Reverse Transcriptase (Invitrogen) according to the manufacturer's instructions. RT-PCR was performed on cDNA from DES$_{water}$ CC samples and corresponding vehicle-treated controls for the genes encoding ERα (*Esr1*), AR (*Ar*), TxA$_2$ receptor (*Tbxa2r*), secreted frizzled related protein 1 (*Sfrp1*), angiopoietin-4 (*Angpt4*), insulin like growth factor binding protein 3 (*Igfbp3*), eNOS (*Nos3*), and oxytocin receptor (*Oxtr*) normalized to the housekeeping gene: ribosomal protein L13a (*Rpl13a*). Significant results from this analysis were repeated with normalization to a second housekeeper ribosomal protein S29 (*Rps29*). RT-PCR was performed on cDNA from DES$_{direct-5}$, Gen$_{direct-20}$ and corresponding vehicle-treated controls for *Esr1*, *Ar*, and *Nos3*, the anti-apoptotic markers B-cell lymphoma-extra large (*BclXL*) and B-cell lymphoma 2 (*Bcl2*), and the apoptotic markers poly (ADP-ribose) polymerase-1 (*Parp1*), Bcl2-antagonist/killer (*Bak1*), and Bcl2-associated X protein (*Bax*). All primers are listed in Supplementary Table 4. The RT-PCR parameters were as follows: 95 °C (2 min.) followed by 40 cycles of 95 °C (15 s) and 57 °C (30 s). Melt curve parameters were 95 °C (5 s), 60 °C (20 s) then 95 °C (1 s).

## Statistics and reproducibility

Prism 9 (version 9.2, Graphpad Software) was used for all statistical analyses. RT-PCR data was generated with the Pfaffl method[168]. To stringently confirm normal distribution, the Anderson-Darling, D'Agostino and Pearson, Shapiro–Wilk, and Kolmogorov–Smirnov tests were used. Only the Shapiro–Wilk and Kolmogorov–Smirnov tests could be applied to sample numbers fewer than $n = 8$. For data sets which did not pass the above normality tests, the Mann–Whitney test was used, except the $DES_{direct-5}$ and $Gen_{-direct-20}$ wire myography data sets which used the Wilcoxon matched-pairs signed rank test. Non-parametric tests such as these are appropriate for analysing non-normal data[169]. Unpaired, two-tailed t-tests used for all other data sets, except the $DES_{direct-5}$ and $Gen_{-direct-20}$ wire myography data sets with normal distributions which used paired, two-tailed t-tests. All continuous data is expressed as the mean ± standard error of the mean (SEM). Data was considered significant when $P < 0.05$. All experiments in the present study are reproducible.

## Reporting summary

Further information on research design is available in the Nature Portfolio Reporting Summary linked to this article.

## Data availability

The authors declare that all the data that support the findings of this study are available within the paper [and the supplementary information file supplementary data 1].

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

## Competing interests

The authors declare no competing interests.
