## [Peer Review File · Communications Biology]

Reviewers' comments:

Reviewer #1 (Remarks to the Author):

The article by Cripps and colleagues describes experiments to test an important hypothesis, namely that estrogenic substances, as representative of endocrine disrupting compounds (EDCs), may in part be responsible for the apparently growing incidence of erectile dysfunction in the human population. They use a mouse model, including long-term feeding from fetal exposure through to adulthood, as well as ex vivo exposures. In terms of methodology, they are using RT-PCR to look at gene expression in the corpora cavernosa (CC) of the adult penis as well as physiological function of the CC ex vivo. The methodology is sound, and the experiments well carried out. Although they are using supraphysiological doses of the model xenoestrogens, DES and genistein, the results are limited, though suggesting strongly that these warrant further study. I have one major comment and several minor amendments to suggest.

Major comment.

Throughout, the authors write that genistein is a xenoestrogen, implying that this is its only mode of action. In fact, genistein is an unusually potent flavonoid, and acts not only as estrogen at estrogen receptors, but is also independently both a potent tyrosine kinase inhibitor (e.g. Barnes et al. 2000; Biofactors 12: 209) as well as a phosphodiesterase inhibitor (e.g. Ko et al., 2004; Biochem Pharmacol 68: 2087). These properties need to be taken into account in the discussion and interpretation of the results. This does not detract from the possibility of EDCs being causal in the context of erectile dysfunction but does broaden the interpretation to other than estrogenic activity.

Minor comments.

Line 60. "are" not "is". Corpora cavernosa is a plural word.

Line 64. "include" not "includes".

Line 73. Delete "to" in "return to".

Line 82. Delete "penis" in "mouse penis".

Line 94. Not "their environment" but "the environment".

Line 102. "phthalates" are not strictly estrogenic. Although their mechanism of action is not yet clear, they definitely do not interact with estrogen receptors, but are considered "anti-androgenic".

Line 117. As above, genistein is not only a xenoestrogen.

Line 310. "were repeated".

Reviewer #2 (Remarks to the Author):

Key results

The authors used a mouse model to evaluate the pharmaceutical DES which had widespread use among a generation of people worldwide. The authors found that DES caused a phenotype consistent with human erectile dysfunction (ED). They found no change in systemic testosterone homeostasis but instead an increase in estrogen receptor Esr1 mRNA expression in mouse erectile tissues. Ex vivo studies of mouse erectile tissues exposed to DES reduced contractility consistent with erectile dysfunction.

Validity

The data are not over-interpreted. The mouse penis is similar to the human penis in its expression of estrogen receptors. The use of phytoestrogen instead of an established control, e.g. estradiol, was disappointing.

Significance

The significance of the study is strong given the widespread use of DES in the middle of the last century and the prevalence of erectile dysfunction, ED. That said, the Introduction was not written in manner to capture this significance. For example, the prevalence of DES use or of fertility issue in

male offspring of DES users is not stated. Further, the paragraph (L117) leading with genistein has nothing to do with ED and reads as off topic despite that it is 'analogous' to DES by way of ER binding and activation.

Data and methodology

Again, I would have liked to see estradiol as a positive control. A reference as to whether C57BL/6 (are these Jackson lab, or other is not described as expected) are an appropriate model of human ED would improve confidence in this study. The high DES dose (which is called 'supraphysiological' despite that DES is not a part of normal physiology) is justified by the authors to reveal the mechanism yet the ER activation by DES has been known for decades and is only superficially evaluated here; it is not a compelling dose choice over the choice of a DES dose that is relevant to the human experience. Further, the exposure of mice to DES post weaning does not reflect the human experience of taking DES during pregnancy. The dose of genistein is also not described with consideration of its intake among various human populations.

Penile outcomes were measured sufficiently well from my view, though I am not an expert in this area of research.

Measuring ER and other mRNA was problematic on numerous fronts. DES is meant to activate ER protein to cause downstream transcription. It is not clear which RNA assessed are transcriptionally regulated by ER transcription factor activity. DES is not widely known to increase ER expression and this change in RNA abundance should have been validated by western blot or other protein quantification methods.

Graphs and legends are clear and easy to interpret.

Analytical approach

The statistical approach was reasonable though it was unclear why four test methods were used for normal distribution analysis, and why non-normal data wasn't transformed.

Suggested improvements

In addition to the above critiques and suggestions, I offer some comments on the discussion.

The complex (biphasic) dose response should be discussed in light of the human exposures that are known for these chemicals.

The fact that it was stated that high DES was used in order to reveal mechanism but at L648 the authors admit they don't understand the mechanism reduces the enthusiasm for this research findings.

Clarity and context

The paragraph at L644 discussing *Esr1* expression does not discuss whether any of the numerous DES studies in rodents have also observed increased *Esr1* as RNA or protein and this should be explicit.

Reviewers' comments:

Reviewer #1 (Remarks to the Author):

The article by Cripps and colleagues describes experiments to test an important hypothesis, namely that estrogenic substances, as representative of endocrine disrupting compounds (EDCs), may in part be responsible for the apparently growing incidence of erectile dysfunction in the human population. They use a mouse model, including long-term feeding from fetal exposure through to adulthood, as well as ex vivo exposures. In terms of methodology, they are using RT-PCR to look at gene expression in the corpora cavernosa (CC) of the adult penis as well as physiological function of the CC ex vivo. The methodology is sound, and the experiments well carried out. Although they are using supraphysiological doses of the model xenoestrogens, DES and genistein, the results are limited, though suggesting strongly that these warrant further study. I have one major comment and several minor amendments to suggest.

Major comment.

Throughout, the authors write that genistein is a xenoestrogen, implying that this is its only mode of action. In fact, genistein is an unusually potent flavonoid, and acts not only as estrogen at estrogen receptors, but is also independently both a potent tyrosine kinase inhibitor (e.g. Barnes et al. 2000; Biofactors 12: 209) as well as a phosphodiesterase inhibitor (e.g. Ko et al., 2004; Biochem Pharmacol 68: 2087). These properties need to be taken into account in the discussion and interpretation of the results. This does not detract from the possibility of EDCs being causal in the context of erectile dysfunction but does broaden the interpretation to other than estrogenic activity.

Minor comments.

Line 60. "are" not "is". Corpora cavernosa is a plural word.

Line 64. "include" not "includes".

Line 73. Delete "to" in "return to".

Line 82. Delete "penis" in "mouse penis".

Line 94. Not "their environment" but "the environment".

Line 102. "phthalates" are not strictly estrogenic. Although their mechanism of action is not yet clear, they definitely do not interact with estrogen receptors, but are considered "anti-androgenic".

Line 117. As above, genistein is not only a xenoestrogen.

Line 310. "were repeated".

Author responses:

Major

We thank the reviewer for their important insight, and we have incorporated the estrogen-independent effects of genistein into the discussion (L: 803-309), as well as briefly in the introduction (L: 138-140). According to the literature, inhibition of tyrosine kinase and the PDEs should augment the relaxation of the rodent CC¹⁻⁴. However, this contrasts with what we observe in the Gen_{direct-20} CC, thus it remains likely that it is the estrogenic effects of genistein which disrupt CC function as observed with DES. These points have again been

incorporated into the discussion so that the audience can gain a broader perspective of these results.

Minor

'Are' is changed to 'is' (plural) (L: 63)

'Includes' changed to 'include' (L: 67)

Deleted 'to' in 'return to' (L: 76)

Deleted 'penis' in 'mouse penis' (L: 85)

Changed 'their environment' to 'the environment' (L: 97)

Phthalates are deleted from the introduction to estrogenic-EDCs (L: 105)

The estrogen-independent pathways of genistein have briefly been introduced (L: 138-140)

'Was repeated' changed to 'were repeated' (L: 353)

Reviewer #2 (Remarks to the Author):

Key results

The authors used a mouse model to evaluate the pharmaceutical DES which had widespread use among a generation of people worldwide. The authors found that DES caused a phenotype consistent with human erectile dysfunction(ED). They found no change in systemic testosterone homeostasis but instead an increase in estrogen receptor Esr1 mRNA expression in mouse erectile tissues. Ex vivo studies of mouse erectile tissues exposed to DES reduced contractility consistent with erectile dysfunction.

Validity

The data are not over-interpreted. The mouse penis is similar to the human penis in its expression of estrogen receptors. The use of phytoestrogen instead of an established control, e.g. estradiol, was disappointing.

Significance

The significance of the study is strong given the widespread use of DES in the middle of the last century and the prevalence of erectile dysfunction, ED. That said, the Introduction was not written in manner to capture this significance. For example, the prevalence of DES use or of fertility issue in male offspring of DES users is not stated. Further, the paragraph (L117) leading with genistein has nothing to do with ED and reads as off topic despite that it is 'analogous' to DES by way of ER binding and activation.

Author responses:

We thank the reviewer for drawing attention to these points. The aim of using DES here was not specifically to establish a link between DES descendants and ED, rather it was used to model widespread exposures to various estrogenic-EDCs experienced by contemporary society. This has been emphasised in the abstract (L: 18-20) the introduction (L: 125-127) and at the beginning of the discussion (L: 653-654). Nonetheless, further information has been included about the widespread historical use of DES and the associated fertility

outcomes in male descendants, thereby providing the audience with a well-known example of the impact of estrogenic-EDCs on human populations (L: 117-122). The paragraph leading with genistein has been substantially altered to stay on the topic of ED by focusing on the studies which have identified that excessive soy consumption as a potential risk factor (L: 129-137). For the same reason, the information regarding a link between DES and ED has also been raised immediately after introducing DES (L: 115-117).

Data and methodology

Again, I would have liked to see estradiol as a positive control. A reference as to whether C57BL/6 (are these Jackson lab, or other is not described as expected) are an appropriate model of human ED would improve confidence in this study. The high DES dose (which is called 'supraphysiological' despite that DES is not a part of normal physiology) is justified by the authors to reveal the mechanism yet the ER activation by DES has been known for decades and is only superficially evaluated here; it is not a compelling dose choice over the choice of a DES dose that is relevant to the human experience. Further, the exposure of mice to DES post weaning does not reflect the human experience of taking DES during pregnancy. The dose of genistein is also not described with consideration of its intake among various human populations.

Penile outcomes were measured sufficiently well from my view, though I am not an expert in this area of research.

Measuring ER and other mRNA was problematic on numerous fronts. DES is meant to activate ER protein to cause downstream transcription. It is not clear which RNA assessed are transcriptionally regulated by ER transcription factor activity. DES is not widely known to increase ER expression and this change in RNA abundance should have been validated by western blot or other protein quantification methods.

Graphs and legends are clear and easy to interpret.

Author responses:

We thank the reviewer for their suggestions. However, estradiol cannot be used as a positive control for DES. In pregnant rodents, estradiol is metabolised rapidly by the placenta and accumulates to a much lesser degree in the fetal tissue compared to DES⁵. DES by virtue of its chemical structure can escape this placental surveillance system and cross the placenta to elicit its detrimental effects on development^{6,7}. In contrast estradiol (commonly used in large quantities during IVF treatments) does not seem to impact fetal development⁸. Given that DES was administered during the window of embryonic development in our study, estradiol could not apply as a positive control in this instance. However, we have edited the methods to briefly communicate the contrasting pharmacokinetics of DES and estradiol in this context (L: 199-201). Furthermore, although estradiol and DES overlap in their mechanisms of action, ER α can elicit distinct transcriptomic signatures and bind to different coregulator motifs depending on its activation by DES or estradiol⁹. Thus, rather than serve as a positive control, the inclusion of estradiol could well elicit its own unique set of transcriptional effects which are beyond the scope of the present study.

We have substituted the term 'supraphysiological' for more appropriate terms pertaining to estrogenic-EDCs (L: 188-190, 238-239, 819). The C57BL/6 mice originated from Jackson

Laboratory which has now been communicated (L: 177). The ED phenotypes of C57BL/6 mice are comparable to that of humans and thus function as an appropriate model for human ED¹⁰⁻¹². In addition, the C57BL/6 background is sensitive to estrogenic endocrine disruption compared to other mouse strains, and thus is suitable for evaluating the impacts of estrogenic-EDCs on humans¹³. These points have now been communicated in the methods (L: 180-182).

We thank the reviewer regarding their comments on our DES and genistein treatments. We recognise that DES is a potent activator of ER. However, it is unclear whether estrogenic-EDCs contribute to ED primarily by a mechanism involving direct activation of ERs localised to the erectile tissue, or indirectly by the suppression of testosterone from the testes, a well-established pathway for these chemicals. It is difficult to disentangle these pathways with smaller doses of DES, thus the high dose serves to clearly define the impacts of estrogenic-EDCs on erectile function. As detailed in our responses to the significance of the study and the respective adjustments to the introduction, the high dose of DES was also used as a model for the broader human experience of exposure to a multitude of estrogenic-EDCs at several potentially sensitive temporal windows. The aim is not to model historic DES exposures during pregnancy, but to demonstrate whether systemic and direct estrogenic-EDC exposures could impact erection physiology, as this is an under-researched field. The above points are emphasised in the abstract (L: 18-20), introduction (L: 125-127), methods (L: 188-193), and in the discussion (L: 651-658). Although the dose of genistein (20 µM) used in the present study can be considered high compared to normal human exposures, it was implemented to first clearly define the effects of genistein on erectile function. These points have been clarified in the methods (L: 238-241) and discussion (L: 651-658). Overall, these doses serve as a starting point to understand the impacts and mechanisms of action of estrogenic-EDC exposure on erectile function, which is part of a sparse group of studies which investigate such a link. We fully expect and hope that future studies will continue the research with different doses, estrogenic-EDCs, and exposure methods. These remarks have also been explicitly stated in the discussion (L: 657-658) and conclusion (L: 883-885). In addition, estrogenic-EDCs typically exhibit non-monotonic effects¹⁴ which is an aspect touched upon in this study (L: 832-838). However, an estrogenic-EDC treatment regime with multiple doses is appropriate for another study entirely given the complex nature of these non-monotonic responses, the full scale of which is beyond the scope here.

We appreciate the reviewer's comments on gene transcription. Of the genes included in our DES_{water} RT-PCR assays, estrogen signalling is known to transcriptionally regulate *Ar*, *Tbxar2*, *Sfrp1*, *Igfbp3* and *Nos3*¹⁵⁻¹⁹. It is unclear whether this is true for *Angpt4* and *Oxtr1*^{20,21}, although they are also implicated in erectile function/ED and were included to screen for any indirect effects of estrogenic-EDCs. These points have been emphasised in the discussion (L: 732-736). Regarding the direct acute estrogenic-EDC exposure RT-PCR assays, previous research has demonstrated that the anti-apoptotic factors *Bcl-XL* and *Bcl2* are transcriptionally regulated by estrogen, as well as the apoptotic factor *Bak1*²²⁻²⁴. It is unclear from prior research whether this is also true of the apoptotic factors *Parp1* and *Bax*^{25,26}. However, these were also included to screen for any indirect cytotoxic effects. This information has been incorporated into the discussion (L: 786-789).

Regarding the increased *Esr1* expression in the CC following systemic DES treatment in our study, at least two other papers have demonstrated that developing rodents treated with DES exhibit increased *Esr1* mRNA and protein within the penis and CC^{27,28}. Thus, we cannot consider our results an isolated anomaly and this has been explicitly stated in the discussion (L: 708-711). However, we have now also communicated that in different tissue types such as the gonads, DES administration does not always affect *Esr1* expression (L: 713-719). Thus, our results point to a potential CC-specific effect of estrogenic-EDCs.

We can appreciate the reviewer's comment on protein quantification of *Esr1*. However, protein quantification is beyond the scope of this experiment as the transcriptional changes we have demonstrated are nonetheless important in interpreting the results, which are also consistent with a prior study which demonstrated increased ER α protein in the rodent CC following DES exposure (described above). In addition, a previous study also showed that endogenous estrogen signalling is a potent mediator of CC contraction²⁹, thus it is logical that increased ER α protein in the DES_{water} CC is responsible for augmented contraction. This point has also been communicated in the discussion (L: 719-721). However, we have also emphasised in the discussion that protein quantification is indeed a cause for future studies (L: 721-723).

Analytical approach

The statistical approach was reasonable though it was unclear why four test methods were used for normal distribution analysis, and why non-normal data wasn't transformed.

Given that the Nature journal guidelines stress that normal and non-normal datasets should be identified, we used four normality tests to distinguish these with stringency. To clarify, t-tests were used for all normal datasets, whereas non-parametric tests (i.e., the Mann-Whitney and Wilcoxon matched-pairs signed rank tests) were used for the non-normal datasets. Previous literature has stated that the use of non-parametric tests are acceptable (and preferable) alternatives to transforming non-normal data³⁰. These points have been communicated in the methods (L: 361-363, 367-368).

Suggested improvements

In addition to the above critiques and suggestions, I offer some comments on the discussion. The complex (biphasic) dose response should be discussed in light of the human exposures that are known for these chemicals.

The fact that it was stated that high DES was used in order to reveal mechanism but at L648 the authors admit they don't understand the mechanism reduces the enthusiasm for this research findings.

We thank the reviewer for drawing attention to what we consider an exciting research area that is the biphasic response of the CC to Acetylcholine (ACh). As it stands however, little is understood about the biphasic mechanism of ACh in the CC, let alone its interaction with estrogenic-EDCs. Thus, it is difficult to expand the discussion on this point beyond explicitly stating that this is an under-researched physiological response and needs to be explored further so that we can better place estrogenic-EDC exposure in the context of human ED (L: 669-674).

Regarding the mechanism, rather than stating we do not understand it, we have rephrased our discussion by clarifying that the molecular signalling apparatus for the impacts of estrogenic-EDCs on CC function requires further investigation (L: 725-727). Our study represents one of the few papers which identifies a cause-and-effect relationship between estrogenic-EDC exposure and increased CC contraction, with a mechanism which likely involves enhanced localised estrogen signalling. We expect and hope that future studies pursue these findings by identifying the genomic or non-genomic effectors for estrogen-mediated CC contraction.

Clarity and context

*The paragraph at L644 discussing *Esr1* expression does not discuss whether any of the numerous DES studies in rodents have also observed increased *Esr1* as RNA or protein and this should be explicit.*

The authors believe this point has been addressed according to the previous suggestions by the reviewer regarding DES. Previous studies demonstrate that DES increases *Esr1* mRNA and protein in the rodent CC (L: 708-711).

- 1 Gur, S., Kadowitz, P. J. & Hellstrom, W. J. A protein tyrosine kinase inhibitor, imatinib mesylate (Gleevec), improves erectile and vascular function secondary to a reduction of hyperglycemia in diabetic rats. *J Sex Med* **7**, 3341-3350, doi:10.1111/j.1743-6109.2010.01922.x (2010).
- 2 Cashen, D. E., MacIntyre, D. E. & Martin, W. J. Effects of sildenafil on erectile activity in mice lacking neuronal or endothelial nitric oxide synthase. *Br J Pharmacol* **136**, 693-700, doi:10.1038/sj.bjp.0704772 (2002).
- 3 Pankey, E. A., Lasker, G. F., Gur, S., Hellstrom, W. J. & Kadowitz, P. J. Analysis of erectile responses to imatinib in the rat. *Urology* **82**, 253.e217-224, doi:10.1016/j.urology.2013.04.009 (2013).
- 4 Bivalacqua, T. J. *et al.* Sildenafil inhibits superoxide formation and prevents endothelial dysfunction in a mouse model of secondhand smoke induced erectile dysfunction. *J Urol* **181**, 899-906, doi:10.1016/j.juro.2008.10.062 (2009).
- 5 Henry, E. C. & Miller, R. K. Comparison of the disposition of diethylstilbestrol and estradiol in the fetal rat. Correlation with teratogenic potency. *Biochem Pharmacol* **35**, 1993-2001, doi:10.1016/0006-2952(86)90732-x (1986).
- 6 Shah, H. C. & McLachlan, J. A. The fate of diethylstilbestrol in the pregnant mouse. *Journal of Pharmacology and Experimental Therapeutics* **197**, 687 (1976).
- 7 Reed, C. E. & Fenton, S. E. Exposure to diethylstilbestrol during sensitive life stages: a legacy of heritable health effects. *Birth defects research. Part C, Embryo today : reviews* **99**, 134-146, doi:10.1002/bdrc.21035 (2013).
- 8 Zhang, J. *et al.* The duration of estrogen treatment before progesterone application does not affect neonatal and perinatal outcomes in frozen embryo transfer cycles. *Front Endocrinol (Lausanne)* **14**, 988398, doi:10.3389/fendo.2023.988398 (2023).
- 9 Adam, A. H. B. *et al.* Estrogen receptor alpha (ER α)-mediated coregulator binding and gene expression discriminates the toxic ER α agonist diethylstilbestrol (DES) from the endogenous ER α agonist 17 β -estradiol (E2). *Cell Biol Toxicol* **36**, 417-435, doi:10.1007/s10565-020-09516-6 (2020).

- 10 Xie, D. *et al.* Mouse model of erectile dysfunction due to diet-induced diabetes mellitus. *Urology* **70**, 196-201, doi:10.1016/j.urology.2007.02.060 (2007).
- 11 Xie, D. *et al.* A mouse model of hypercholesterolemia-induced erectile dysfunction. *J Sex Med* **4**, 898-907, doi:10.1111/j.1743-6109.2007.00518.x (2007).
- 12 Musicki, B. *et al.* Hypercholesterolemia-induced erectile dysfunction: endothelial nitric oxide synthase (eNOS) uncoupling in the mouse penis by NAD(P)H oxidase. *J Sex Med* **7**, 3023-3032, doi:10.1111/j.1743-6109.2010.01880.x (2010).
- 13 Spearow, J. L., Doemeny, P., Sera, R., Leffler, R. & Barkley, M. Genetic Variation in Susceptibility to Endocrine Disruption by Estrogen in Mice. *Science* **285**, 1259-1261 (1999).
- 14 Hill, C. E., Myers, J. P. & Vandenberg, L. N. Nonmonotonic Dose-Response Curves Occur in Dose Ranges That Are Relevant to Regulatory Decision-Making. *Dose Response* **16**, 1559325818798282, doi:10.1177/1559325818798282 (2018).
- 15 Cripps, S. M., Mattiske, D. M. & Pask, A. J. Erectile Dysfunction in Men on the Rise: Is There a Link with Endocrine Disrupting Chemicals? *Sex Dev* **15**, 187-212, doi:10.1159/000516600 (2021).
- 16 Gregory, K. J. & Schneider, S. S. Estrogen-mediated signaling is differentially affected by the expression levels of Sfrp1 in mammary epithelial cells. *Cell Biol Int* **39**, 873-879, doi:10.1002/cbin.10468 (2015).
- 17 Huynh, H., Yang, X. & Pollak, M. Estradiol and antiestrogens regulate a growth inhibitory insulin-like growth factor binding protein 3 autocrine loop in human breast cancer cells. *J Biol Chem* **271**, 1016-1021, doi:10.1074/jbc.271.2.1016 (1996).
- 18 McAbee, M. D. & DonCarlos, L. L. Estrogen, But Not Androgens, Regulates Androgen Receptor Messenger Ribonucleic Acid Expression in the Developing Male Rat Forebrain*. *Endocrinology* **140**, 3674-3681, doi:10.1210/endo.140.8.6901 (1999).
- 19 Nott, S. L. *et al.* Genomic responses from the estrogen-responsive element-dependent signaling pathway mediated by estrogen receptor alpha are required to elicit cellular alterations. *J Biol Chem* **284**, 15277-15288, doi:10.1074/jbc.M900365200 (2009).
- 20 Gimpl, G. & Fahrenholz, F. The Oxytocin Receptor System: Structure, Function, and Regulation. *Physiological Reviews* **81**, 629-683, doi:10.1152/physrev.2001.81.2.629 (2001).
- 21 Currie, M. J. *et al.* Angiopoietin-1 Is Inversely Related to Thymidine Phosphorylase Expression in Human Breast Cancer, Indicating a Role in Vascular Remodeling¹. *Clinical Cancer Research* **7**, 918-927 (2001).
- 22 Pike, C. J. Estrogen Modulates Neuronal Bcl-xl Expression and β -Amyloid-Induced Apoptosis. *Journal of Neurochemistry* **72**, 1552-1563, doi:<https://doi.org/10.1046/j.1471-4159.1999.721552.x> (1999).
- 23 Dong, L. *et al.* Mechanisms of Transcriptional Activation of bcl-2 Gene Expression by 17 β -Estradiol in Breast Cancer Cells*. *Journal of Biological Chemistry* **274**, 32099-32107, doi:<https://doi.org/10.1074/jbc.274.45.32099> (1999).
- 24 Leung, L. K., Do, L. & Wang, T. T. Regulation of death promoter Bak expression by cell density and 17 beta-estradiol in MCF-7 cells. *Cancer Lett* **124**, 47-52, doi:10.1016/s0304-3835(97)00430-8 (1998).
- 25 Gadad, S. S. *et al.* PARP-1 Regulates Estrogen-Dependent Gene Expression in Estrogen Receptor α -Positive Breast Cancer Cells. *Mol Cancer Res* **19**, 1688-1698, doi:10.1158/1541-7786.Mcr-21-0103 (2021).

- 26 Choi, K.-C., Kang, S. K., Tai, C.-J., Auersperg, N. & Leung, P. C. K. Estradiol Up-Regulates Antiapoptotic Bcl-2 Messenger Ribonucleic Acid and Protein in Tumorigenic Ovarian Surface Epithelium Cells*. *Endocrinology* **142**, 2351-2360, doi:10.1210/endo.142.6.8144 (2001).
- 27 Okumu, L. A., Brinton, S., Braden, T. D., Simon, L. & Goyal, H. O. Estrogen-induced maldevelopment of the penis involves down-regulation of myosin heavy chain 11 (MYH11) expression, a biomarker for smooth muscle cell differentiation. *Biology of reproduction* **87**, 109-109, doi:10.1095/biolreprod.112.103556 (2012).
- 28 Goyal, H. O. *et al.* Abnormal Morphology of the Penis in Male Rats Exposed Neonatally to Diethylstilbestrol Is Associated with Altered Profile of Estrogen Receptor- α Protein, but Not of Androgen Receptor Protein: A Developmental and Immunocytochemical Study1. *Biology of Reproduction* **70**, 1504-1517, doi:10.1095/biolreprod.103.026328 (2004).
- 29 Vignozzi, L. *et al.* Oxytocin receptor is expressed in the penis and mediates an estrogen-dependent smooth muscle contractility. *Endocrinology* **145**, 1823-1834, doi:10.1210/en.2003-0962 (2004).
- 30 Kühnast, C. & Neuhäuser, M. A note on the use of the non-parametric Wilcoxon-Mann-Whitney test in the analysis of medical studies. *Ger Med Sci* **6**, Doc02 (2008).

REVIEWERS' COMMENTS:

Reviewer #1 (Remarks to the Author):

The comments and criticisms of this reviewer have been adequately addressed in the revised manuscript.

Reviewer #2 (Remarks to the Author):

The authors have addressed all previous comments nearly to my satisfaction. Specifically they have provided better justification of their selection of a positive control, which is an imperfect selection choice for complex traits due to multiple mechanisms often at play.

An issue remains with the discussion of biphasic response. It is known that exogenous/synthetic estrogenic chemicals can have other biological targets. This and receptor saturation are often evoked to explain non-monotonic relationships. A simple search of acetylcholine and estrogen on pubmed yields nearly 1000 publications. Given Communications Biology articles should bring new biological insights to its readers, one sentence could speculate what likely mechanisms could be operating in this context by analogy to other tissues.

Although I would have liked to see ERalpha protein presented, I do not feel the absence of protein abundance merits the exclusion of this body of research from impactful excellence and the authors now address this limitation.